# PERTURB TO FORGET: ZERO-SHOT MACHINE UN-LEARNING

## ABSTRACT

Machine unlearning seeks to remove the influence of specific data from trained models, a requirement increasingly critical under modern privacy regulations. Yet most existing approaches either depend on costly retraining or require access to the original dataset, which may be unavailable or restricted. We propose Inversion-Guided Neuron Perturbation (IGNP), a zero-shot framework that performs unlearning entirely without the original data. IGNP begins by synthesizing class-representative samples through a model inversion-inspired process, enabling analysis of how different parameters encode forget and retain classes. By contrasting these sensitivities, IGNP identifies parameters that are especially critical for encoding the forget class, while being less influential for retain classes. This strategy erases targeted knowledge with precision while preserving model utility. Extensive experiments on multiple benchmarks demonstrate that IGNP achieves complete forgetting with minimal accuracy loss, outperforms state-of-the-art zero-shot and data-dependent baselines, and provides strong resistance to membership inference and inversion attacks. These results establish IGNP as a practical and efficient solution for data-free unlearning in compliance-driven machine learning.

## 1 INTRODUCTION

Increasingly stringent data privacy regulations, such as the EU's General Data Protection Regulation (GDPR) and its "Right to be Forgotten"(Voigt & Von dem Bussche, 2017), are driving the need to remove specific data from trained machine learning models. These models, often trained on vast datasets containing sensitive or copyrighted information, must comply with legal requirements granting individuals the right to have their data erased. This process, known as machine unlearning, aims to eliminate the influence of targeted data points while maintaining model performance, posing significant research challenges in balancing effectiveness and efficiency.

The concept of machine unlearning emerged from the need to address limitations of complete model retraining. While retraining a model from scratch on an amended dataset guarantees exact unlearning by ensuring the updated model is statistically indistinguishable from a newly trained one (Guo et al., 2020), its prohibitive cost is infeasible for large-scale applications (Bourtoule et al., 2021). Early research aimed to optimize this process with structured methods like Sharded, Isolated, Sliced, and Aggregated (SISA) training, which partitions data to avoid a full retraining cycle by only retraining affected sub-models (Bourtoule et al., 2021). However, these exact methods can still incur significant overhead and may not always perform well, highlighting the tension between perfect removal and practical efficiency (Triantafillou et al., 2024). This challenge directly motivated developing approximate unlearning techniques.

However, many prominent approximate unlearning methods introduce a critical flaw: a dependency on the original training data. Some approaches, for instance, require access to retained data samples for a fine-tuning step (Tarun et al., 2024; Zuo et al., 2025), which creates both computational overhead and data storage burdens. Other techniques circumvent this by pre-calculating data-dependent information so the raw data can be discarded (Foster et al., 2024), but this is impractical for legacy models where the original data is already unavailable. Furthermore, even ostensibly data-free methods can incur high computational costs and perform poorly on larger datasets (Chundawat et al., 2023a). This fundamental reliance on sensitive data contradicts the principle of data minimization and severely restricts the practical deployment of these methods in secure, real-world environments.

To address these challenges, we introduce Inversion-Guided Neuron Perturbation (IGNP), a novel zero-shot, data-free unlearning framework that operates without access to original training samples, whether forgotten or retained. IGNP employs a three-stage process: first, it generates synthetic class-representative data samples using a model inversion-like method, dividing them into forget and retain sets; second, it constructs a sensitivity information matrix by measuring the model's parameter sensitivity to both sets; and finally, it applies a binary search to determine an optimal threshold for perturbing the most sensitive parameters, effectively nullifying target information while minimizing impact on model utility, thus providing a privacy-compliant and practical solution for model maintenance.

Our work makes the following key contributions: ❶ We propose IGNP, a novel framework that introduces a data-free approach to zero-shot machine unlearning by generating class-representative samples through a model inversion process, eliminating the need for access to the original training dataset while maintaining accuracy and efficiency. ❷ An adaptive perturbation mechanism is designed to selectively weaken parameters tied to the forget class, where synthesized data guide a binary search calibration and fine-grained scaling ensures effective forgetting without damaging the utility of retained knowledge. ❸ Comprehensive evaluations across multiple benchmarks demonstrate IGNP's superiority over existing methods, achieving complete and precise forgetting with substantially reduced computational cost, and highlighting its practicality for privacy-compliant machine learning.

## 2 RELATED WORK

**Machine Unlearning**

Recent machine unlearning methods aim to efficiently remove the influence of specific data from trained models, avoiding costly retraining. One approach utilizes teacher-student frameworks for targeted forgetting (Chundawat et al., 2023b), which has been adapted for zero-shot scenarios (Chundawat et al., 2023a). Another line of research leverages the Fisher Information Matrix (FIM) to guide the unlearning process by identifying parameters crucial to the data being removed. While early FIM-based methods were computationally expensive (Golatkar et al., 2020), Selective Synaptic Dampening (SSD) improves efficiency by using the FIM's diagonal to suppress influential parameters related to the forget-set (Foster et al., 2024). Other techniques include fine-grained parameter perturbation (Zuo et al., 2025) and zero-glance methods that inject error-maximizing noise to erase class-specific knowledge (Tarun et al., 2024). However, many existing techniques face challenges such as high computational or storage overhead (Graves et al., 2021) or require access to original training data, which can introduce privacy risks.

**Model Parameter Sensitivity Evaluation** Parameter sensitivity analysis identifies critical model parameters by evaluating how changes affect the model's loss, often interpreted through the loss function's second-order derivative (Maltoni & Lomonaco, 2019). While the Hessian matrix was an early tool for this purpose (Le Cun et al., 1989), its computation is infeasible for modern neural networks. The Fisher Information Matrix (FIM) has emerged as a practical alternative for quantifying parameter importance (Kirkpatrick et al., 2017; Guo et al., 2020), with its diagonal approximation being widely used in fields like pruning and continual learning (Singh & Alistarh, 2020). This analysis is particularly relevant to unlearning because over-parameterized models are prone to memorizing training data (Feldman, 2020; Carlini et al., 2019). A key limitation, however, is that these methods often assume local quadraticity or require original training data, restricting their applicability in data-free scenarios.

**Privacy Attacks on Machine Unlearning Models** The objective of machine unlearning is to produce an updated model whose output distribution is statistically indistinguishable from a model never trained on the forgotten data, a goal closely aligned with differential privacy (Dwork & Roth, 2014; Ginart et al., 2019). This goal is directly challenged by privacy attacks that aim to extract sensitive information. Key threats include model inversion attacks, which reconstruct training data from model predictions (Fredrikson et al., 2015), and membership inference attacks (MIAs), which determine if a specific data point was in the training set (Shokri et al., 2017; Hu et al., 2022). MIAs often work by exploiting the model's higher confidence on training data, sometimes using shadow models to learn this behavior. Consequently, the robustness of an unlearning method against such attacks has become a critical benchmark for evaluating its effectiveness.

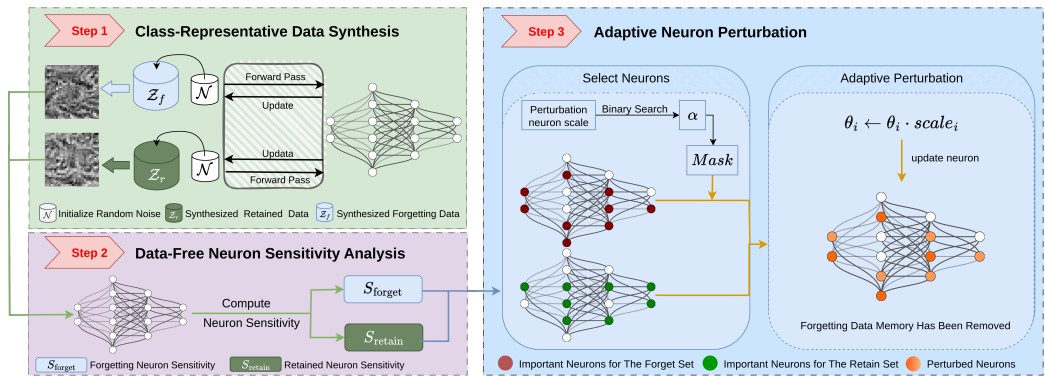

Figure 1: The process of IGNP unfolds in three stages. First, an initial set of random noise tensors, $\mathcal{N}$, is directly optimized into representative samples for the forget ($\mathcal{Z}_f$) and retain ($\mathcal{Z}_r$) classes via a model inversion-inspired procedure. Second, these samples are leveraged to compute parameter sensitivity, which allows for the identification of parameters critical to the forget class, marked by a binary $Mask$. Finally, a controlled-magnitude perturbation, guided by an adaptive binary search, is applied to the identified parameters to erase forget-class memories and preserve overall model performance.

## 3 METHOD

### 3.1 PROBLEM FORMULATION AND METHOD OVERVIEW

Let the complete training dataset be denoted by $\mathcal{D} = \{(x_i, y_i)\}_{i=1}^N$, where $x_i \in \mathcal{X} \subseteq \mathbb{R}^d$ is an input data sample and $y_i \in \mathcal{Y} = \{1, \ldots, K\}$ is its corresponding label from one of $K$ classes. In the context of machine unlearning, this dataset is conceptually partitioned into two disjoint subsets: the forget set, $\mathcal{D}_f$, containing samples to be erased, and the retain set, $\mathcal{D}_r$, comprising the remaining data, satisfying $\mathcal{D} = \mathcal{D}_f \cup \mathcal{D}_r$ and $\mathcal{D}_f \cap \mathcal{D}_r = \emptyset$. Let an original model $M_\theta$ be trained on $\mathcal{D}$. The ideal outcome of unlearning is defined by a retrained model, $M_\theta^r$, which is a model trained from scratch exclusively on the retain set $\mathcal{D}_r$. The objective of an unlearning algorithm is to efficiently produce an unlearned model, $M_\theta'$, that emulates the behavior of this retrained model without the high computational cost of full retraining.

In this work, we introduce Inversion-Guided Neuron Perturbation (IGNP), a framework designed to efficiently and precisely erase the influence of a target forget class, $\mathcal{C}_f$, from a pre-trained model while preserving the utility of the retain classes, $\mathcal{C}_r$. Crucially, IGNP operates without requiring access to the original training data. By leveraging model inversion-inspired class-representative data to approximate the target data, identifying critical parameters encoding the forget class, and applying controlled perturbations, our method addresses the privacy, scalability, and computational challenges of existing unlearning methods. The framework operates in three sequential stages: (1) Class-Representative Data Synthesis, (2) Data-Free Neuron Sensitivity Analysis, and (3) Adaptive Neuron Perturbation. The overall process is illustrated in Figure 1 and Algorithm 1.

### 3.2 CLASS-REPRESENTATIVE DATA SYNTHESIS

To eliminate reliance on original training data, IGNP employs a technique inspired by model inversion to synthesize class-representative samples(Fredrikson et al., 2015). Our key insight is that for the purpose of analyzing parameter importance, a small set of highly representative synthetic samples is sufficient, removing the need for the full dataset. The synthesis process is an iterative optimization procedure based on gradient descent. Instead of updating the model's weights, we treat the input tensors themselves as trainable parameters. The process begins by drawing a batch of random noise tensors, $\mathcal{N} = \{z_1, \ldots, z_n\}$, from a Gaussian distribution. These tensors are then iteratively updated over a series of optimization steps to minimize a composite loss function. This procedure effectively transforms the initial random noise into class prototypes that strongly activate the model's neurons for a target class.

The optimization is guided by a composite loss function designed to achieve two goals: ensuring the samples are representative of the target class and maintaining diversity among them. The first objective is achieved using a standard cross-entropy (CE) loss, which pushes the model's prediction for each synthetic sample $z_i$ towards the target class label $y_{\text{target}}$. To prevent the samples from collapsing to a single point (mode collapse) and encourage them to represent different facets of the class, we introduce a diversity regularizer. This term maximizes the average pairwise L2 distance between the vectorized synthetic samples. We denote the vectorized representation of a sample $z_i$ as $\mathbf{z}_i$ and define the average distance as:

$$\bar{d}(\mathcal{N}) = \frac{1}{\binom{n}{2}} \sum_{1 \leq i < j \leq n} \|\mathbf{z}_i - \mathbf{z}_j\|_2 \tag{1}$$

The total loss, $\mathcal{L}$, used to optimize the noise set $\mathcal{N}$ towards a class $y_{\text{target}}$ is then formulated as:

$$\mathcal{L} = \left( \frac{1}{n} \sum_{i=1}^{n} \text{CE}(M(z_i), y_{\text{target}}) \right) + \lambda_{\text{div}} \cdot \left( \bar{d}(\mathcal{N}) \right)^{-1} \tag{2}$$

Here, $M(z_i)$ represents the model's output logits for a synthetic sample $z_i$, CE is the Cross-Entropy loss, and $\lambda_{\text{div}}$ is a weighting hyperparameter. We use the inverse of the average distance, $\left( \bar{d}(\mathcal{N}) \right)^{-1}$, in the loss function because standard optimizers perform minimization. Minimizing the inverse of the distance is equivalent to maximizing the distance itself.

This optimization procedure is applied independently to generate compact sets of synthetic samples: $\mathcal{Z}_f$ for the forget class and $\mathcal{Z}_r$ for the retain classes, by optimizing initial noise sets against their respective target labels. While the resulting samples may not be perfectly discernible to the human eye, they are highly efficient to generate and potent enough to capture the core class-level features learned by the model. These proxies are thus ideally suited for the subsequent Data-Free Neuron Sensitivity Analysis.

### 3.3 DATA-FREE NEURON SENSITIVITY ANALYSIS

Leveraging the synthetic samples $\mathcal{Z}_f$ and $\mathcal{Z}_r$, IGNP assesses the importance of each model parameter in encoding information for both the forget and retain classes. To achieve this, we establish separate sensitivity profiles for each parameter $\theta_i$ within the target layers. Adopting the metric proposed in(Aljundi et al., 2018), the sensitivity with respect to the forget class, denoted as $S_{\text{forget}}(\theta_i)$, is computed over the synthetic forget samples $\mathcal{Z}_f$ as the average squared gradient of the loss:

$$S_{\text{forget}}(\theta_i) = \frac{1}{N_{\text{forget}}} \sum_{k=1}^{N_{\text{forget}}} \left( \frac{\partial \mathcal{L}(f(x_k; \theta), y_k)}{\partial \theta_i} \right)^2 \tag{3}$$

Similarly, the sensitivity with respect to the retain classes, $S_{\text{retain}}(\theta_i)$, is calculated using the synthetic retain samples $\mathcal{Z}_r$:

$$S_{\text{retain}}(\theta_i) = \frac{1}{N_{\text{retain}}} \sum_{k=1}^{N_{\text{retain}}} \left( \frac{\partial \mathcal{L}(f(x_k; \theta), y_k)}{\partial \theta_i} \right)^2 \tag{4}$$

where $f(x_k; \theta)$ is the model's output for a synthetic sample $x_k$, $\mathcal{L}$ is the cross-entropy loss, while $N_{\text{forget}}$ and $N_{\text{retain}}$ are the number of samples in the sets $\mathcal{Z}_f$ and $\mathcal{Z}_r$, respectively.

Calculating these two sensitivity profiles provides a stable measure of each parameter's influence. This allows us to identify which parameters are most critical to the forget class relative to the retain classes, enabling targeted intervention without reliance on original data, as emphasized in our zero-shot and data-free objectives.

### 3.4 ADAPTIVE NEURON PERTURBATION

The final stage of IGNP selectively modifies model parameters to eliminate the influence of the forget class while preserving performance on the retain classes. We construct this phase based on the

parameter dampening foundation of Selective Synaptic Dampening (SSD) (Foster et al., 2024) and optimize it specifically for the zero-shot unlearning scenario. Unlike SSD, which relies on real data to yield comparable gradient magnitudes, IGNP must operate on synthesized pseudo-samples. This shift introduces a critical scale mismatch problem, where gradient norms fluctuate unpredictably, rendering standard data-dependent methods unstable. This shift introduces unique challenges regarding gradient stability and scale, necessitating a fine-grained, adaptive perturbation strategy centered on two core principles: identifying parameters that are disproportionately critical to the forget class under noisy approximation, and attenuating their influence in a calibration-aware manner.

First, we pinpoint parameters that are most influential for the forget class relative to the retain classes. A fundamental challenge in using synthetic data is scale mismatch. Since sensitivity matrices are derived from optimized noise rather than i.i.d. real data, the gradient norms for the forget and retain sets often differ by orders of magnitude. This stochasticity makes the absolute sensitivity scores incomparable, causing fixed-threshold methods (like those used in SSD) to fail. To address this, we introduce an adaptive mask based on a calibrated threshold, $\alpha$, which acts as a dynamic scaling factor to align the differing magnitudes of the two sensitivity profiles for a relative comparison:

$$\text{Mask}_i = \begin{cases} 1 & \text{if } S_{\text{forget},i} > \alpha \cdot S_{\text{retain},i} \\ 0 & \text{otherwise} \end{cases} \tag{5}$$

Determining the optimal value for $\alpha$ is non-trivial in a data-free scenario. Our empirical analysis reveals that the numerical scale of the optimal $\alpha$ fluctuates dramatically across different models and datasets (ranging from $10^{-5}$ to $10^5$) due to the variability of synthetic noise. Consequently, treating $\alpha$ as a static hyperparameter is infeasible. However, we observe that while the gradient scales fluctuate, the physical proportion of parameters requiring modification for effective forgetting remains remarkably consistent. Leveraging this stability, we employ an automated calibration method to decouple the unlearning performance from numerical noise. Instead of searching for the elusive optimal $\alpha$ directly, we pre-specify a target coverage range (e.g., $[C_{\min}, C_{\max}]$, typically 1%–5% of total parameters). We then use a binary search algorithm to dynamically find the precise $\alpha$ that satisfies:

$$\sum_i \text{Mask}_i(\alpha) \in [N \cdot C_{\min}, N \cdot C_{\max}] \tag{6}$$

where $N$ is the total number of parameters. This transforms the unstable hyperparameter selection into a robust constraint satisfaction problem.

Once the forget-critical parameters are identified, we attenuate their values using a tailored scaling factor. The design of this factor specifically addresses the scale mismatch inherent to synthetic data. It is defined as:

$$\text{scale}_i = \begin{cases} \frac{\alpha}{\lambda} \cdot \frac{S_{\text{retain},i}}{S_{\text{forget},i}} & \text{if } \text{Mask}_i = 1 \\ 1 & \text{otherwise} \end{cases} \tag{7}$$

The term $\frac{S_{\text{retain},i}}{S_{\text{forget},i}}$ provides sensitivity-aware scaling. However, due to the stochastic volatility of synthetic data, $S_{\text{forget},i}$ may be significantly smaller than $S_{\text{retain},i}$ depending on the noise optimization trajectory. Such scale mismatch causes the raw ratio to become arbitrarily large, potentially exceeding 1 and leading to erroneous parameter enhancement. To address this, we explicitly incorporate $\alpha$ as a *normalization term* within the modulation factor $\frac{\alpha}{\lambda}$. Since $\alpha$ is dynamically derived to bridge the gap between $S_{\text{forget}}$ and $S_{\text{retain}}$ (as defined in Eq. 5), its inclusion here mathematically cancels out the global scale mismatch. This ensures that the final scaling factor is strictly bounded within a valid dampening range ($< 1$), preventing model collapse regardless of the fluctuating gradient norms.

The parameter update is executed in a single efficient step: $\theta_i' \leftarrow \theta_i \cdot \text{scale}_i$. This mechanism ensures that the perturbation is proportional to the relative importance of the parameter, effectively erasing target information. By integrating data synthesis with this adaptive calibration mechanism, IGNP achieves precise, privacy-preserving unlearning without requiring access to the original training data.

---

**Algorithm 1** Inversion-Guided Neuron Perturbation

---

**Input:** $M_\theta$ (model parameters), $C_f$ (forget class)
**Parameter:** $\lambda$ (perturbation strength), $[C_{\min}, C_{\max}]$ (perturbation scale range)
**Output:** $M'_\theta$ (perturbed model parameters)
 1: Class-Representative Data Synthesis: $\mathcal{Z}_f$, $\mathcal{Z}_r$
 2: Compute sensitivities $S_{\text{forget},i}$ and $S_{\text{retain},i}$
 3: Use binary search to find the threshold parameter $\alpha$ such that the mask coverage $\mathcal{C}(\alpha)$ satisfies $\mathcal{C}(\alpha) \in [C_{\min}, C_{\max}]$
 4: **for** each $\theta_i \in M_\theta$ **do**
 5: $\quad$ $\text{Mask}_i \leftarrow S_{\text{forget},i} > \alpha \cdot S_{\text{retain},i}$
 6: $\quad$ $scale_i \leftarrow \begin{cases} \frac{\alpha}{\lambda} \cdot \frac{S_{\text{retain},i}}{S_{\text{forget},i}} & \text{if } \text{Mask}_i = 1, \\ 1 & \text{otherwise} \end{cases}$
 7: $\quad$ $\theta_i \leftarrow \theta_i \cdot scale_i$
 8: **end for**

---

## 4 EXPERIMENTAL SETUP

**Datasets and Models** We evaluated our method on four image classification benchmarks: MNIST (Lecun et al., 1998), SVHN (Netzer et al., 2011), CIFAR-10, and CIFAR-100 (Krizhevsky & Hinton, 2009). To assess scalability, we used three CNN architectures of varying complexity: a lightweight LeNet (Lecun et al., 1998), a 9-layer ResNet-9, and a deeper ResNet-18 (He et al., 2016). The ResNet-18 was specifically adapted for the $32 \times 32$ resolution of the CIFAR datasets.

**Training Details** All models were trained from scratch using a batch size of 128, with convolutional and linear layers initialized via Kaiming normal initialization (He et al., 2015). Our primary training configuration utilized the SGD optimizer with a momentum of 0.9 and weight decay of $5 \times 10^{-4}$. Models were typically trained for 200 epochs with a cosine annealing learning rate schedule, starting from an initial rate of 0.01. Specific modifications, such as employing the Adam optimizer for the LeNet architecture or using label smoothing for CIFAR-100, were made where appropriate. For all experiments, we saved the model checkpoint with the highest validation accuracy. Further granular details for each experimental setup are provided in the appendix.

**Zero-Shot Class Unlearning Setting** Our work tackles the challenging task of Zero-Shot Class Unlearning (Li et al., 2025; Chundawat et al., 2023a), a stringent yet practical setting where the unlearning algorithm is denied access to any original training data, from either the forget set ($\mathcal{D}_f$) or the retain set ($\mathcal{D}_r$). This constraint emulates real-world scenarios governed by strict privacy laws that mandate permanent data deletion. Our proposed method, alongside the GKT baseline, operates under this strict zero-shot condition. We assess performance in two key scenarios: single-class unlearning and sequential multi-class unlearning, where the model must forget additional classes in succession. For a comprehensive evaluation, we also compare against baseline methods that relax this constraint by requiring access to the retain set ($\mathcal{D}_r$).

**Baseline Methods** To evaluate the efficacy of our proposed approach, we conduct a comparative analysis against a representative set of baselines. The gold-standard Retrain serves as the theoretical upper bound, involving training a new model from scratch on the retain set ($D_r$) for 200 epochs using the same training protocol as the original model, albeit at significant computational cost. Additionally, we include Finetune, which further trains the original pre-trained model on the retain set ($D_r$) for 20 epochs to reinforce performance on non-target classes while attempting to diminish the influence of the forget set ($D_f$). We also compare against several state-of-the-art unlearning methods: Selective Synaptic Dampening (SSD) (Foster et al., 2024), which leverages the Fisher Information Matrix to select high-importance parameters for dampening UNSIR (Tarun et al., 2024), which employs an "impair-repair" framework; and the Gated Knowledge Transfer (GKT) scheme (Chundawat et al., 2023a), a prominent method that also operates under the zero-shot unlearning constraint.

**Evaluation Metrics** We evaluate the unlearning process across three key dimensions: efficacy, privacy, and efficiency. Efficacy is assessed using Retain Set Accuracy, to ensure performance on non-target data is preserved, and Forget Set Accuracy, to confirm the erasure of target information. For privacy, we employ Membership Inference Attacks (Chundawat et al., 2023b) to detect potential

information leakage. Finally, efficiency is measured by the Unlearning Time to quantify computational overhead. For a fair comparison, hyperparameters for all baseline methods were carefully tuned according to their original specifications.

**IGNP parameters** For our proposed IGNP method, we set the trade-off parameter $\lambda$ to 10 across all experiments to balance the learning objectives. The perturbation scale was configured based on the dataset's complexity. For the 10-class datasets, SVHN, CIFAR-10, and MNIST, this value was set within the range of [0.040, 0.050]. For the more complex CIFAR-100 dataset, this value was set within the range of [0.011, 0.013].

## 5 Experiment Evaluations

### 5.1 Single-Class Unlearning Effectiveness

The primary goal of an effective unlearning method is to eliminate targeted information while maintaining the model's overall utility. We assess this dual objective by evaluating accuracy on the forget set ($\mathcal{D}_f$), which should ideally approach random chance, and the retain set ($\mathcal{D}_r$), which should remain comparable to the original model's performance. Detailed results for single-class unlearning are presented in Table 1.

Table 1: Performance of LeNet and ResNet on MNIST, SVHN, CIFAR-10, and CIFAR-100. $D_r$ and $D_f$ denote the classification accuracy (in %) on the retained and forgotten data, respectively. For all datasets, the results correspond to forgetting the first class.

| Model | Dataset | Acc. | Original | Retrain | Finetune | GKT | SSD | UNSIR | Ours |
|---|---|---|---|---|---|---|---|---|---|
| LeNet | MNIST | $\mathcal{D}_r \uparrow$ | 99.30 | 99.21 | 99.31 | 97.95 | 99.16 | 97.69 | **99.35** |
| | | $\mathcal{D}_f \downarrow$ | 99.80 | 0.00 | 0.00 | **0.00** | **0.00** | **0.00** | **0.00** |
| | SVHN | $\mathcal{D}_r \uparrow$ | 91.55 | 90.90 | 91.36 | 78.17 | 88.53 | 58.05 | **90.68** |
| | | $\mathcal{D}_f \downarrow$ | 91.23 | 0.00 | 0.00 | **0.00** | **0.00** | **0.00** | **0.00** |
| | CIFAR10 | $\mathcal{D}_r \uparrow$ | 73.47 | 76.74 | 74.86 | 24.26 | 73.27 | 39.19 | **73.60** |
| | | $\mathcal{D}_f \downarrow$ | 79.20 | 0.00 | 0.00 | **0.00** | **0.00** | **0.00** | **0.00** |
| ResNet9 | MNIST | $\mathcal{D}_r \uparrow$ | 99.58 | 99.60 | 99.58 | 94.10 | **95.84** | 90.19 | 94.14 |
| | | $\mathcal{D}_f \downarrow$ | 99.90 | 0.00 | 99.80 | **0.00** | **0.00** | **0.00** | **0.00** |
| | SVHN | $\mathcal{D}_r \uparrow$ | 95.56 | 95.95 | 95.71 | 87.52 | 86.04 | 84.49 | **94.80** |
| | | $\mathcal{D}_f \downarrow$ | 96.27 | 0.00 | 90.65 | **0.00** | **0.00** | **0.00** | **0.00** |
| | CIFAR10 | $\mathcal{D}_r \uparrow$ | 91.06 | 92.52 | 91.59 | 14.29 | 86.82 | 58.70 | **86.88** |
| | | $\mathcal{D}_f \downarrow$ | 92.80 | 0.00 | 69.70 | **0.00** | **0.00** | **0.00** | **0.00** |
| | CIFAR100 | $\mathcal{D}_r \uparrow$ | 73.88 | 73.97 | 73.57 | 1.90 | 71.64 | 34.06 | **73.94** |
| | | $\mathcal{D}_f \downarrow$ | 89.00 | 0.00 | 35.00 | **0.00** | **0.00** | **0.00** | **0.00** |
| ResNet18 | CIFAR10 | $\mathcal{D}_r \uparrow$ | 93.78 | 93.34 | 93.86 | 21.09 | 92.48 | 49.02 | **93.42** |
| | | $\mathcal{D}_f \downarrow$ | 94.70 | 0.00 | 88.60 | **0.00** | **0.00** | **0.00** | **0.00** |
| | CIFAR100 | $\mathcal{D}_r \uparrow$ | 77.38 | 75.68 | 77.86 | 1.78 | 76.53 | 32.45 | **76.80** |
| | | $\mathcal{D}_f \downarrow$ | 87.00 | 0.00 | 61.00 | **0.00** | **0.00** | **0.00** | **0.00** |

As shown in Table 1, our method, IGNP, consistently achieves complete unlearning, reducing forget set accuracy ($\mathcal{D}_f$) to 0.00% across all settings, matching the ideal Retrain baseline. The key differentiator among methods is the preservation of retain set accuracy ($\mathcal{D}_r$), where IGNP demonstrates superior performance. For instance, on the challenging ResNet18-CIFAR100 task, IGNP's retain accuracy (76.80%) is minimally affected compared to the Original model (77.38%) and even surpasses the Retrain baseline (75.68%). In contrast, competing methods show significant flaws. Finetune often fails to forget adequately (e.g., 61.00% $\mathcal{D}_f$ on ResNet18-CIFAR100), while others like GKT and UNSIR can cause a catastrophic collapse in performance on the retain set (e.g., 14.29% and 58.70% respectively on ResNet9-CIFAR10). While SSD presents competitive results, IGNP generally maintains higher fidelity to the original model's performance across most scenarios.

These results highlight IGNP's ability to achieve a superior trade-off, thoroughly removing targeted information with minimal impact on valuable retained knowledge.

## 5.2 CONTINUAL UNLEARNING EFFECTIVENESS

In the challenging continual unlearning scenario, where classes are forgotten sequentially, IGNP demonstrates superior robustness and stability (Table 2). This is most evident on ResNet18-CIFAR10: after the second unlearning step, methods like Finetune and SSD effectively fail to forget ($\mathcal{D}_f > 95\%$). In stark contrast, IGNP maintains perfect forgetting (0.00% $\mathcal{D}_f$) and high utility on the retain set (92.62% $\mathcal{D}_r$). This stability is consistent, with retain accuracy on SVHN dropping by only 0.15% after two steps, proving IGNP's efficacy for sequential unlearning.

Table 2: Continual unlearning results for LeNet on MNIST, ResNet9 on SVHN, and ResNet18 on CIFAR-10, compared across different unlearning methods. The Retrain and Finetune methods perform a one-shot removal of all forgotten classes, while other methods perform sequential unlearning, building upon the previous unlearning step.

| Model | Step | Original | Retrain | Finetune | GKT | SSD | UNSIR | Ours |
|---|---|---|---|---|---|---|---|---|
| LeNet-MNIST | 0: $\mathcal{D}_r \uparrow$ | 99.30 | 99.21 | 99.31 | 97.95 | 99.16 | 97.69 | **99.35** |
| | 0: $\mathcal{D}_f \downarrow$ | 99.80 | 0.00 | 0.00 | **0.00** | **0.00** | **0.00** | **0.00** |
| | 1 (after 0): $\mathcal{D}_r \uparrow$ | 99.09 | 99.18 | 99.24 | 97.45 | 94.75 | 98.01 | **99.06** |
| | 1 (after 0): $\mathcal{D}_f \downarrow$ | 99.65 | 0.00 | 0.00 | **0.00** | **0.00** | **0.00** | **0.00** |
| ResNet9-SVHN | 0: $\mathcal{D}_r \uparrow$ | 95.56 | 95.95 | 95.71 | 87.52 | 86.04 | 84.49 | **94.80** |
| | 0: $\mathcal{D}_f \downarrow$ | 96.27 | 0.00 | 90.65 | **0.00** | **0.00** | **0.00** | **0.00** |
| | 1 (after 0): $\mathcal{D}_r \uparrow$ | 94.03 | 95.56 | 95.77 | 83.17 | 81.99 | 88.94 | **94.64** |
| | 1 (after 0): $\mathcal{D}_f \downarrow$ | 97.67 | 0.00 | 92.90 | **0.00** | **0.00** | **0.00** | **0.00** |
| ResNet18-CIFAR10 | 0: $\mathcal{D}_r \uparrow$ | 93.78 | 93.34 | 93.86 | 21.09 | 92.48 | 49.02 | **93.42** |
| | 0: $\mathcal{D}_f \downarrow$ | 94.70 | 0.00 | 88.60 | **0.00** | **0.00** | **0.00** | **0.00** |
| | 1 (after 0): $\mathcal{D}_r \uparrow$ | 93.20 | 91.14 | 93.61 | 12.50 | **92.91** | 28.82 | 92.62 |
| | 1 (after 0): $\mathcal{D}_f \downarrow$ | 97.00 | 0.00 | 95.10 | **0.00** | 98.10 | **0.00** | **0.00** |

## 5.3 EFFICIENCY OF UNLEARNING

Unlearning efficiency is critical for practical applications. As shown in Table 3, our IGNP framework demonstrates exceptional performance. Compared to the zero-shot competitor GKT, IGNP requires only 12% of the time. It is also nearly two orders of magnitude faster than retraining from scratch ($<1\%$ of the time) while achieving a comparable outcome. Although data-dependent methods like UNSIR are slightly faster, IGNP offers a highly competitive timescale without needing the original data, making it ideal for timely and private data removal in real-world scenarios.

Table 3: Statistics of time cost and zero-shot ability for different methods on the ResNet18 model, forgetting the class airplane in the CIFAR-10 dataset.

| Metric | Retrain | Finetune | UNSIR | SSD | GKT | Ours |
|---|---|---|---|---|---|---|
| Zero-shot | No | No | No | No | Yes | Yes |
| Time Cost (s) | 4287 | 230 | 12 | 16 | 285 | 34 |

## 5.4 MEMBERSHIP INFERENCE ATTACKS

To evaluate IGNP's privacy protection, we tested its resilience against Membership Inference Attacks (MIAs), which aim to determine if a data point was part of a model's training set. As presented in Table 4, IGNP demonstrates robust defense capabilities, consistently reducing MIA scores

to levels comparable with, and often superior to, the Retrain-from-scratch baseline. For instance, with ResNet-18 on CIFAR-10, the attack score plummets from 90.86% to a near-zero 0.02%, significantly outperforming the 18.48% achieved by retraining. This stands in contrast to the unreliable Finetune baseline, which in some cases even increased the model's vulnerability. These findings confirm that IGNP effectively erases discernible traces of the forgotten data, achieving a level of privacy protection that meets or exceeds that of complete model retraining.

Table 4: Membership Inference Attack Efficacy (in %).

| Model | Dataset | Original | Retrain | Finetune | GKT | SSD | UNSIR | Ours |
|-------|---------|----------|---------|----------|-----|-----|-------|------|
| LeNet | MNIST | 93.25 | 7.67 | 3.71 | 14.38 | 0.22 | 9.76 | **0.68** |
| | SVHN | 79.33 | 30.05 | 13.62 | 12.23 | 12.41 | 63.34 | **10.31** |
| | CIFAR10 | 44.46 | 16.80 | 69.62 | 49.18 | 52.94 | 41.38 | **0.00** |
| ResNet9 | MNIST | 94.07 | 0.10 | 90.92 | 9.26 | 10.61 | 13.11 | **8.66** |
| | SVHN | 85.61 | 1.37 | 53.27 | **3.96** | 3.54 | 12.81 | 17.46 |
| | CIFAR10 | 87.54 | 14.16 | 34.28 | 65.20 | 21.20 | 45.28 | **20.48** |
| | CIFAR100 | 79.00 | 17.40 | 4.40 | 53.80 | 79.40 | 48.96 | **1.80** |
| ResNet18 | CIFAR10 | 90.86 | 18.48 | 38.96 | 85.22 | 0.12 | 17.18 | **0.02** |
| | CIFAR100 | 84.60 | 21.20 | 4.60 | 47.60 | 20.00 | 69.80 | **2.40** |

## 5.5 ABLATION STUDY

To validate our selective perturbation module, we conduct an ablation study by replacing it with a naive zeroing strategy. This strategy nullifies the top-k% most influential parameters for the forget class instead of attenuating them. As shown in Table 5, this substitution is highly detrimental. Zeroing a mere 1% of parameters causes a catastrophic 11.84% drop in retain accuracy, indicating these parameters are crucial for shared knowledge. In stark contrast, our full framework achieves complete forgetting while reducing retain accuracy by only a negligible 0.37%. This underscores that precisely attenuating parameters, rather than crudely eliminating them, is essential for preserving shared knowledge while effectively unlearning the target class.

Table 5: Ablation on removing binary search: directly zeroing top-$k$% parameters most related to the forgotten class (CIFAR-10, ResNet-18). Reported are averages over 10 classes.

| Variant | Retain (init→final) | Forget (init→final) | Retain $\Delta$ | Forget $\Delta$ |
|---------|---------------------|---------------------|----------|----------|
| 1% | 93.87 → 82.03 | 93.87 → 15.07 | -11.84 | -78.80 |
| 3% | 93.87 → 70.88 | 93.87 → 3.59 | -22.99 | -90.28 |
| 5% | 93.87 → 59.65 | 93.87 → 0.03 | -34.22 | -93.84 |
| Ours | 93.87 → 93.50 | 93.87 → 0.00 | -0.37 | -93.87 |

## 6 CONCLUSION

We propose Inversion-Guided Neuron Perturbation (IGNP), a zero-shot, data-free machine unlearning framework. Without requiring original data, IGNP approximates target information and adaptively perturbs knowledge-encoding parameters to erase it. Experiments demonstrate that IGNP achieves superior unlearning accuracy, computational efficiency, and resilience to privacy attacks, all without retraining. Future work will focus on scaling IGNP to large-scale datasets and further streamlining the process by obtaining parameter sensitivity directly during noise generation.

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

# A APPENDIX

## A.1 THE USE OF LARGE LANGUAGE MODELS

The language of this paper was polished using large language models (LLMs) to enhance clarity and readability. The final content and academic integrity remain the responsibility of the authors.

## A.2 ORIGINAL MODEL TRAINING HYPERPARAMETERS

Table 6 presents the training hyperparameters used for different model and dataset combinations in our experiments. We carefully selected optimizer types, learning rates, and scheduling approaches based on model architecture and dataset complexity to ensure optimal baseline performance. All experiments were conducted on a system with a 12th Gen Intel(R) Core(TM) i9-12900K CPU, 64GB of RAM, and an NVIDIA RTX 3090 GPU, running Ubuntu 22.04.1 LTS with Python 3.12 and PyTorch 2.7.1.

Table 6: Training Hyperparameters for Model and Dataset Combinations

| Model Dataset | Optimizer | LR | LR Schedule | Momentum | Weight Decay | Epochs |
|---|---|---|---|---|---|---|
| LeNet(Any) | Adam | 0.001 | ReduceLROnPlateau[1] | N/A | 5e-4 | 200 |
| ResNet9(CIFAR10) | SGD | 0.01 | CosineAnnealing | 0.9 | 5e-4 | 200 |
| ResNet9(SVHN) | SGD | 0.01 | CosineAnnealing | 0.9 | 5e-4 | 200 |
| ResNet9(MNIST) | SGD | 0.01 | CosineAnnealing | 0.9 | 5e-4 | 200 |
| ResNet18(CIFAR10) | SGD | 0.01 | CosineAnnealing | 0.9 | 5e-4 | 200 |
| ResNet18(CIFAR100) | SGD | 0.1 | Warmup-Cosine[2] | 0.9 | 1e-3[3] | 200 |

**Notes:**
[1] LeNet's learning rate is reduced by a factor of 0.1 if validation accuracy does not improve for 5 epochs.
[2] For ResNet18 on CIFAR-100, a 5-epoch linear warmup is used, followed by a cosine decay schedule.
[3] For the CIFAR-100 dataset, label smoothing with a factor of 0.1 is applied to the Cross-Entropy loss.

## A.3 MODEL ARCHITECTURES AND TARGET LAYERS

This section details the specific architectures of the models used in our experiments and identifies the layers targeted for perturbation by our proposed unlearning method. Table 7 provides a summary of the layer configurations for LeNet, ResNet9, and ResNet18. Table 8 lists the specific layers that were automatically selected for modification during the unlearning process.

Table 7: Architectures of Models Used in Experiments

| Model | Layer Name | Layer Configuration |
|-------|-----------|---------------------|
| LeNet | conv1 | Conv2d(6, 5x5), ReLU, MaxPool(2x2) |
| | conv2 | Conv2d(16, 5x5), ReLU, MaxPool(2x2) |
| | fc | Linear(400, 120), ReLU, Linear(120, 84), ReLU, Linear(84, C) |
| ResNet9 | conv1 | Conv2d(64, 3x3), BatchNorm, ReLU |
| | conv2 | Conv2d(128, 3x3), BatchNorm, ReLU, MaxPool(2x2) |
| | res1 | BasicBlock(128, 128) |
| | conv3 | Conv2d(256, 3x3), BatchNorm, ReLU, MaxPool(2x2) |
| | conv4 | Conv2d(512, 3x3), BatchNorm, ReLU, MaxPool(2x2) |
| | res2 | BasicBlock(512, 512) |
| | classifier | Linear(512, C) |
| ResNet18 | conv1 | Conv2d(64, 7x7, stride=2), BatchNorm, ReLU, MaxPool(3x3, stride=2) |
| | layer1 | 2 x BasicBlock(64, 64) |
| | layer2 | 2 x BasicBlock(128, 128, stride=2) |
| | layer3 | 2 x BasicBlock(256, 256, stride=2) |
| | layer4 | 2 x BasicBlock(512, 512, stride=2) |
| | fc | Linear(512, C) |
| CifarResNet18 | conv1 | Conv2d(64, 3x3), BatchNorm, ReLU |
| | layer1 | 2 x BasicBlock(64, 64) |
| | layer2 | 2 x BasicBlock(128, 128, stride=2) |
| | layer3 | 2 x BasicBlock(256, 256, stride=2) |
| | layer4 | 2 x BasicBlock(512, 512, stride=2) |
| | fc | Linear(512, C) |

Table 8: Target Layers for Perturbation by Our Method

| Model | Target Layers |
|-------|---------------|
| LeNet | conv2, fc |
| ResNet9 | res1, conv3, res2, conv4, classifier |
| ResNet18 / CifarResNet18 | layer3, layer4, fc |

## A.4 IMPACT OF SYNTHETIC DATA TRAINING EPOCHS AND QUANTITY ON EXPERIMENTAL RESULTS

Figure 2 demonstrates the effect of varying noise sample quantities and training epochs on the unlearning process. The visualization illustrates the critical balance between noise injection and training duration in machine unlearning scenarios for ResNet18 trained on CIFAR-10 dataset.

Our analysis reveals that the number of noise samples (ranging from 100 to 600) significantly impacts unlearning effectiveness, with the results emphasizing the necessity of halting training at optimal early epochs. Sufficient noise enables rapid target data removal while preserving overall model performance on test data. The proper calibration prevents degradation of the model's general classification accuracy, providing valuable insights for optimizing unlearning strategies to achieve efficient target data removal without compromising model utility.Based on these findings, we ultimately choose 200 noise samples and 200 optimization epochs.

## A.5 MODEL INVERSION ATTACK RESISTANCE

Figures 3presents the results of model inversion attacks across different model states. A model inversion attack attempts to reconstruct a representative sample of a class from the trained model's

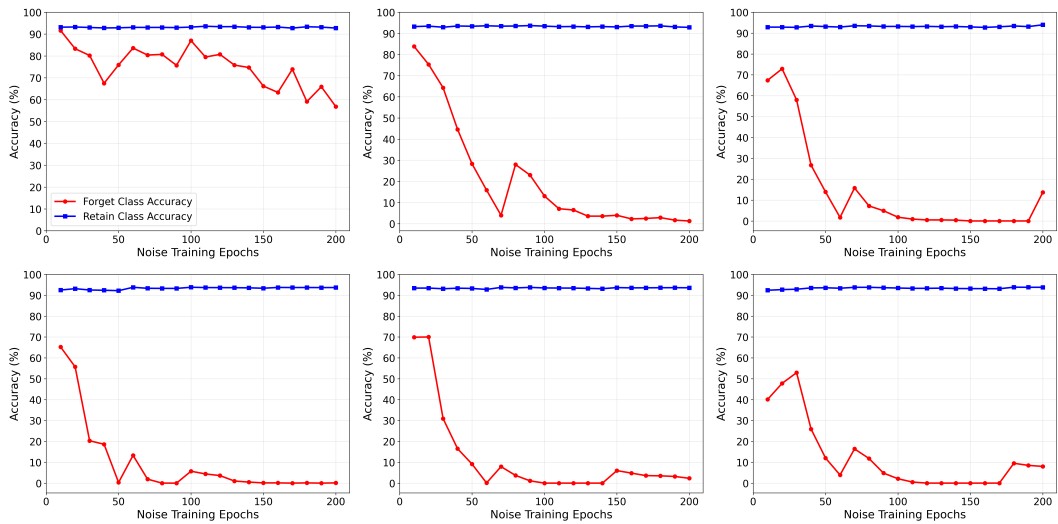

Figure 2: Effect of Noise Sample Quantities and Training Epochs on Unlearning Performance

parameters, where a successful attack on a forgotten class would imply that significant conceptual information about that class still resides within the model. The results demonstrate clear evidence

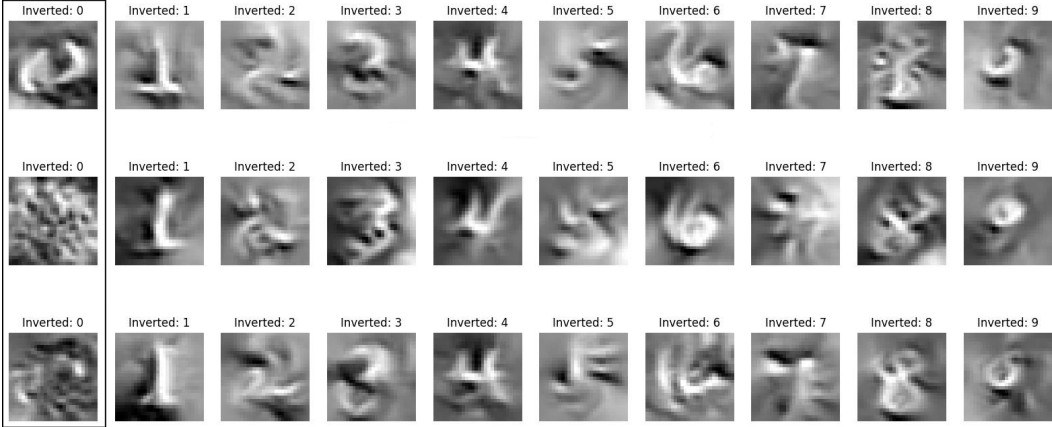

Figure 3: Inversion Attack

of our method's effectiveness across three model states: the original trained model, our unlearned model after forgetting class "0", and the retrained baseline model. In the original model, all digit classes including class "0" can be successfully reconstructed with recognizable visual features, indicating that class-specific information is clearly retained in the model parameters. However, our unlearned model shows a stark contrast where other digit classes remain reconstructible while class "0" produces only random noise with no discernible structure, demonstrating effective erasure of the target class information. The retrained model shows similar reconstruction capabilities to the original model for retained classes, serving as the gold standard for comparison. This comprehensive evaluation provides strong evidence that our unlearning method successfully eliminates latent features specific to the forgotten class while preserving the model's ability to represent other classes, indicating robust protection against model inversion attacks.

## A.6 Impact of Different Lambda Settings on Experimental Results

The parameter $\lambda$ plays a crucial role in balancing unlearning effectiveness and model performance retention. We conducted comprehensive experiments to analyze how different $\lambda$ values affect the unlearning process across various model-dataset combinations.

Our experimental analysis across nine model-dataset combinations (Figure 4) reveals consistent patterns in how $\lambda$ values influence unlearning performance. The forget class accuracy exhibits a rapid declining trend as $\lambda$ values increase across all combinations. When $\lambda = 1$, forget class accuracy remains relatively high (typically > 50%) in some cases, indicating insufficient unlearning effectiveness, while $\lambda \geq 3$ causes forget class accuracy to drop sharply to near 0%, demonstrating effective erasure of target class information.

Retain class accuracy demonstrates remarkable stability throughout the $\lambda$ value variations, maintaining levels close to the original performance in most cases and indicating good preservation of overall model functionality. Some complex datasets like CIFAR-100 show slight performance degradation at larger $\lambda$ values, but these remain within acceptable ranges. The experimental results also reveal interesting differences across model architectures, where LeNet series exhibits the most stable performance on simple datasets with minimal impact from $\lambda$ value changes, ResNet9 series demonstrates excellent balance across various datasets with clear unlearning effects while maintaining stable retention performance, and ResNet18 series requires more precise $\lambda$ value tuning on complex datasets but achieves excellent overall performance.

Dataset complexity significantly influences the optimal $\lambda$ value selection. Simple datasets like MNIST allow for a wider $\lambda$ value selection range with good results achievable from $\lambda = 3 - 20$, medium complexity datasets including CIFAR-10 and SVHN typically require optimal $\lambda$ values between 5-15, while complex datasets such as CIFAR-100 demand more careful $\lambda$ value selection as excessively large $\lambda$ values may impact retain class performance.

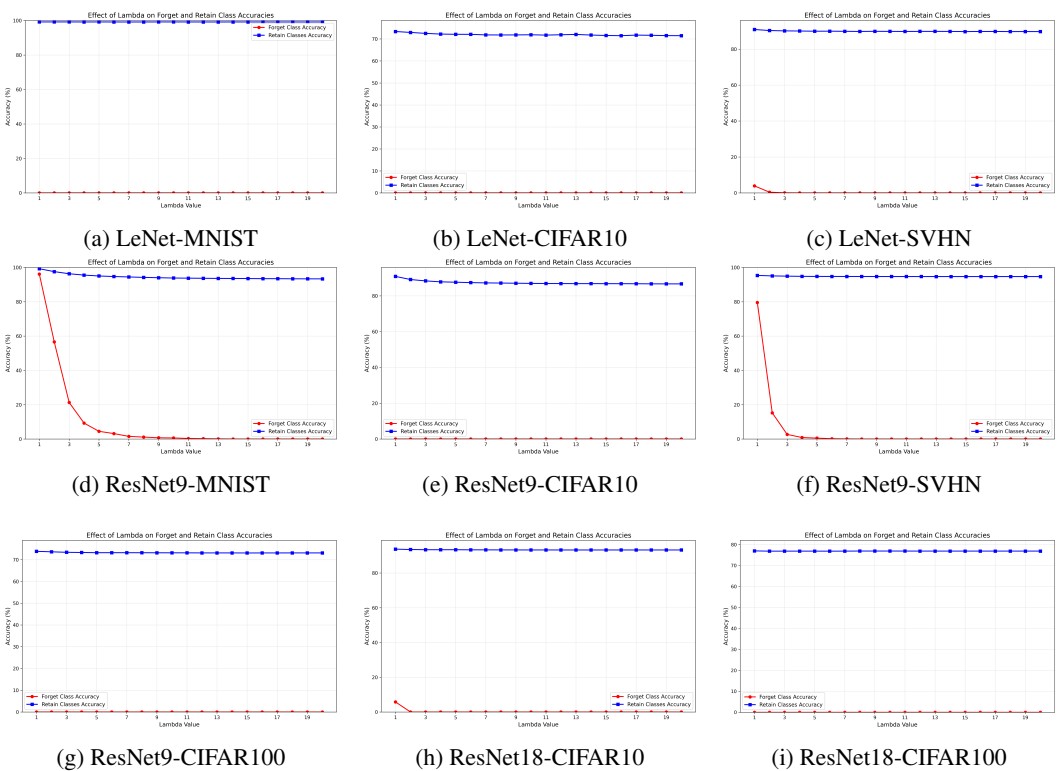

Figure 4: Lambda Parameter Analysis Across Model-Dataset Combinations

After comprehensive analysis of all experimental results, we selected $\lambda = 10$ as the unified regularization parameter. This choice ensures unlearning effectiveness by successfully reducing forget class accuracy to below 1% across all model-dataset combinations while providing more thorough unlearning effects compared to smaller values. The parameter also maintains retention performance balance, keeping retain class accuracy at high levels across all experiments while avoiding excessive impact on retain class performance compared to larger values.

The selection of $\lambda = 10$ demonstrates good consistency across different model architectures and various datasets, which simplifies practical application of the method by avoiding complex parameter tuning for different scenarios. This value is positioned at the center of the effective range, providing good robustness to parameter fine-tuning and maintaining stable performance even under different experimental environments or data distributions. Additionally, the moderate $\lambda$ value ensures stable convergence during the training process while avoiding potential gradient vanishing or training instability issues that may arise from excessively large $\lambda$ values.

### A.7 Impact of Perturbation Parameter Coverage on Experimental Results

The coverage parameter determines the proportion of model parameters that undergo perturbation during the unlearning process. This parameter is crucial for controlling the scope of model modification and directly affects both unlearning effectiveness and retention performance.

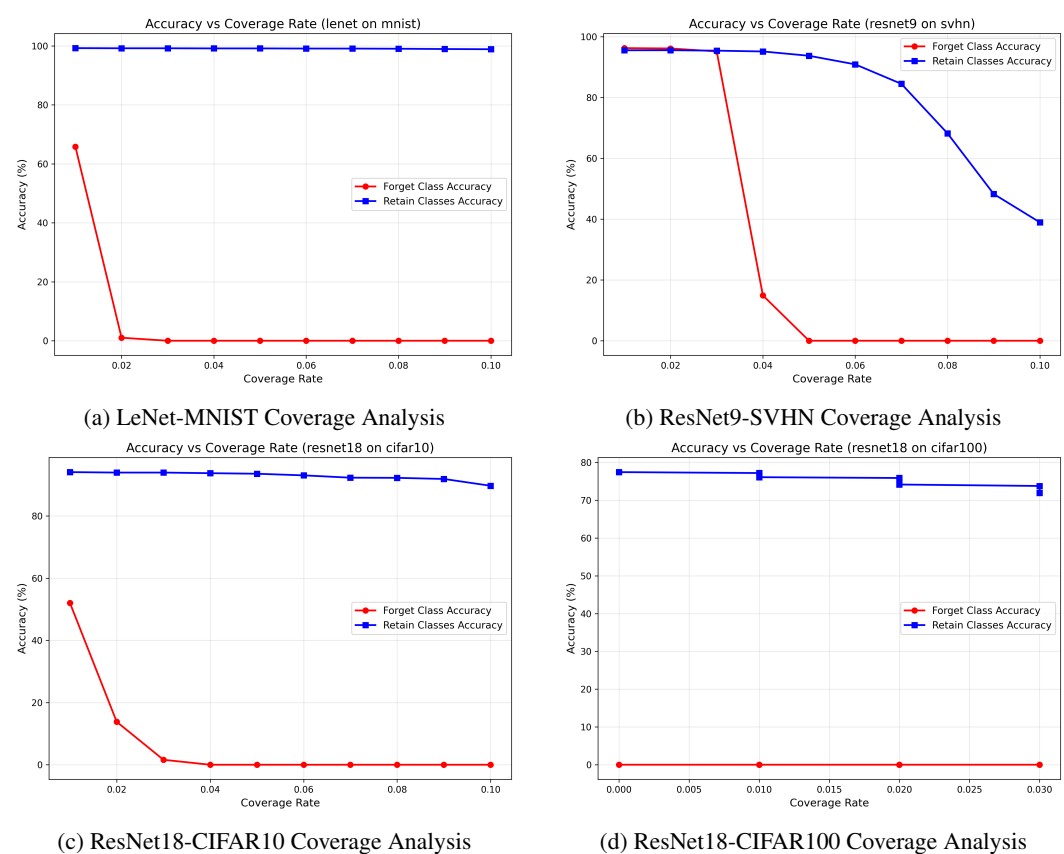

(a) LeNet-MNIST Coverage Analysis   (b) ResNet9-SVHN Coverage Analysis

(c) ResNet18-CIFAR10 Coverage Analysis   (d) ResNet18-CIFAR100 Coverage Analysis

Figure 5: Coverage Parameter Analysis Across Model-Dataset Combinations

Our analysis across different model-dataset combinations (Figure 5) reveals distinct patterns in how coverage parameters influence unlearning performance. ResNet9 on SVHN dataset shows excellent unlearning performance with forget class accuracy dropping from approximately 96% at coverage rate 0.01 to nearly 0% at coverage rate 0.04, while maintain remarkable stability in retain class accuracy at approximately 96% across all coverage rates with minimal degradation. LeNet on MNIST dataset exhibits sharp decline in forget class accuracy from approximately 66% at coverage rate 0.01

to 1% at coverage rate 0.02, maintaining exceptional stability at 99% retain class accuracy across the entire coverage range.

ResNet18 demonstrates different sensitivity patterns depending on the dataset complexity. On CIFAR-100, forget class accuracy remains consistently near 0% across all tested coverage rates, indicating effective unlearning even at very low coverage values, though retain class accuracy shows slight but steady decline from 77% to 72% as coverage increases. On CIFAR-10, forget class accuracy shows dramatic improvement from 52% at coverage rate 0.01 to 14% at 0.02, continuing to decline toward 0% by coverage rate 0.04, while retain class accuracy gradually declines from 94% to 87% but remains within acceptable ranges.

The comparison across different dataset complexities reveals that simple datasets like MNIST show the highest tolerance to coverage parameter variations with minimal impact on retain class accuracy, medium complexity datasets including CIFAR-10 and SVHN exhibit balanced sensitivity with clear optimal operating ranges, while complex datasets such as CIFAR-100 display the highest sensitivity to coverage changes and require conservative parameter selection. Architecture-specific behavior analysis shows that LeNet networks demonstrate the most robust performance with wide acceptable coverage ranges, ResNet9 networks show excellent stability with moderate coverage requirements, and ResNet18 networks require more precise tuning, particularly on complex datasets.

Based on comprehensive experimental analysis, we adopted a task-specific coverage selection strategy that accounts for dataset complexity and model architecture characteristics. For CIFAR-100 datasets, we use coverage ranges of [0.011, 0.013] because complex datasets require conservative parameter modification to preserve intricate feature representations, with fine-tuning within this narrow range ensuring effective unlearning while maintaining classification accuracy on numerous classes.

For 10-class datasets including MNIST, CIFAR-10, and SVHN, we employ coverage ranges of [0.040, 0.050] since simpler classification tasks allow for more aggressive parameter perturbation, with higher coverage values ensuring complete erasure of target class information while maintaining robust performance on remaining classes. Our adaptive selection methodology involves initial coverage estimation based on dataset complexity and model architecture, followed by fine-tuning through grid search within identified optimal ranges and validation using both unlearning effectiveness and retention performance metrics.This systematic approach to coverage determination ensures optimal unlearning performance across diverse experimental settings while maintaining the practical applicability of our proposed IGSP method. The analysis demonstrates that our method's effectiveness depends critically on appropriate parameter selection tailored to specific task characteristics.

## A.8 EXPERIMENTS ON TINY IMAGENET

To assess the efficacy of IGNP on more complex and challenging datasets, we performed single-class unlearning experiments on Tiny ImageNet using a ResNet18 backbone. Tiny ImageNet, with its 200 classes and higher-resolution images, presents a more demanding scenario for evaluating an unlearning method's ability to preserve model utility compared to CIFAR-10/100.

The model was fine-tuned on Tiny ImageNet using a pretrained ResNet18. The detailed training configuration is summarized in Table 9.

The comprehensive results, detailed in Table 10, confirm the consistent effectiveness of our approach. For every class targeted for unlearning, the forget accuracy was successfully reduced to 0.00%, demonstrating complete removal. Concurrently, the impact on the retain set accuracy remained minimal, with performance degradation observed in the narrow range of 0.61% to 1.51%. Furthermore, the unlearning process maintained high efficiency, modifying fewer than 0.5% of the model's parameters in each case. These findings highlight the method's precision in selectively erasing information while preserving the model's knowledge of the remaining classes.

These results confirm that our proposed method, IGNP, remains highly effective even on more complex, large-scale datasets like Tiny ImageNet. It successfully removes the target class information while preserving the model's performance on the remaining data, showcasing its scalability and utility for real-world applications.

Table 9: Training settings for ResNet18 on Tiny ImageNet.

| Parameter | Value |
|---|---|
| Model | ResNet18 (pretrained on ImageNet) |
| Dataset | Tiny ImageNet |
| Optimizer | SGD |
| Initial Learning Rate | 0.01 |
| Momentum | 0.9 |
| Weight Decay | 1e-3 |
| Batch Size | 128 |
| Epochs | 100 |
| Scheduler | Warmup Cosine Annealing |
| Loss Function | Cross-Entropy with Label Smoothing (0.1) |

Table 10: Single-class unlearning results for ResNet18 on Tiny ImageNet.

| Forgotten Class | Forget Accuracy (%) | | Retain Accuracy (%) | | Retain Acc. Drop (%) | Modified Params (%) |
|---|---|---|---|---|---|---|
| | Before | After | Before | After | | |
| class_0 | 82.00 | 0.00 | 55.91 | 54.40 | 1.51 | 0.42 |
| class_10 | 38.00 | 0.00 | 56.13 | 54.65 | 1.48 | 0.44 |
| class_20 | 68.00 | 0.00 | 55.98 | 55.37 | 0.61 | 0.44 |
| class_30 | 38.00 | 0.00 | 56.13 | 54.98 | 1.15 | 0.44 |

## A.9 EXPERIMENTS ON THE VISION TRANSFORMER MODEL

To further validate the versatility of our proposed method, we extended our evaluation to the Vision Transformer (ViT) architecture, a paradigm fundamentally different from CNNs. We employed a ViT model, pretrained on ImageNet and fine-tuned on the CIFAR-10 dataset. The unlearning task was to forget the 'airplane' class. For this experiment, our method was configured to target the final layers of the model, specifically blocks.10, blocks.11, norm, and head.

The unlearning results are summarized in Table 11.

Table 11: Single-class unlearning results for ViT-Tiny on CIFAR-10.

| Metric | Before Unlearning | After Unlearning |
|---|---|---|
| Forget Class ('airplane') Accuracy | 98.50% | 0.50% |
| Retain Set Accuracy | 97.31% | 91.06% |
| Accuracy Drop (Retain) | 6.26% | |

The experiment demonstrates that our method successfully generalizes to Transformer-based models. The acc. for the 'airplane' class was effectively nullified, dropping by 98.00%. While the perf. on the retain set saw a more noticeable decline of 6.26% compared to the ResNet experiments, this is expected given the highly interconnected nature of the attn. mechanism in ViT models.

## A.10 MEAN AND STANDARD DEVIATION OVER MULTIPLE RUNS

To further strengthen the reliability of our experimental results, we conducted three independent runs with different random seeds for all main experiments and report the mean and standard deviation (Mean ± Std) of both the retention set accuracy and the forgotten set accuracy. Table 12–17 summarize the results for our method and all baselines. Across all datasets and architectures, our method achieves near-zero forgotten-set accuracy with competitive or superior retention-set performance. Notably, both our method and GKT are zero-sample unlearning methods that do not require access to the original training data during unlearning, yet our method consistently attains substan-

tially higher retention accuracy than GKT while maintaining the same strong forgetting performance (0.00% on the forgotten set).

Table 12: Mean ± Std of accuracy (%) over three runs for our method.

| Dataset | Model | Retention Set Result | Forgotten Set Result |
|---|---|---|---|
| MNIST | LeNet | 99.16±0.02 | 0.00±0.00 |
| SVHN | LeNet | 90.09±0.30 | 0.00±0.00 |
| CIFAR-10 | LeNet | 71.96±0.18 | 0.00±0.00 |
| MNIST | ResNet9 | 93.44±0.66 | 0.00±0.00 |
| SVHN | ResNet9 | 94.45±0.72 | 0.00±0.00 |
| CIFAR-10 | ResNet9 | 85.24±1.53 | 0.00±0.00 |
| CIFAR-100 | ResNet9 | 73.36±0.17 | 0.00±0.00 |
| CIFAR-10 | ResNet18 | 93.33±0.15 | 0.00±0.00 |
| CIFAR-100 | ResNet18 | 76.59±0.16 | 0.00±0.00 |

Table 13: Mean ± Std of accuracy (%) over three runs for the Retrain baseline.

| Dataset | Model | Retention Set Result | Forgotten Set Result |
|---|---|---|---|
| MNIST | LeNet | 99.28±0.08 | 0.00±0.00 |
| SVHN | LeNet | 90.75±0.28 | 0.00±0.00 |
| CIFAR-10 | LeNet | 76.52±0.76 | 0.00±0.00 |
| MNIST | ResNet9 | 99.53±0.07 | 0.00±0.00 |
| SVHN | ResNet9 | 95.30±0.18 | 0.00±0.00 |
| CIFAR-10 | ResNet9 | 89.83±0.29 | 0.00±0.00 |
| CIFAR-100 | ResNet9 | 72.32±0.40 | 0.00±0.00 |
| CIFAR-10 | ResNet18 | 91.53±0.24 | 0.00±0.00 |
| CIFAR-100 | ResNet18 | 73.46±0.30 | 0.00±0.00 |

Table 14: Mean ± Std of accuracy (%) over three runs for the Finetune baseline.

| Dataset | Model | Retention Set Result | Forgotten Set Result |
|---|---|---|---|
| MNIST | LeNet | 99.32±0.04 | 0.00±0.00 |
| SVHN | LeNet | 91.32±0.08 | 0.00±0.00 |
| CIFAR-10 | LeNet | 75.21±0.14 | 0.00±0.00 |
| MNIST | ResNet9 | 99.58±0.02 | 99.90±0.00 |
| SVHN | ResNet9 | 95.68±0.03 | 90.48±0.21 |
| CIFAR-10 | ResNet9 | 91.69±0.11 | 69.87±0.21 |
| CIFAR-100 | ResNet9 | 73.64±0.07 | 35.67±1.53 |
| CIFAR-10 | ResNet18 | 93.83±0.07 | 88.47±0.58 |
| CIFAR-100 | ResNet18 | 77.61±0.09 | 57.33±2.52 |

## A.11 PER-CLASS RETAIN ACCURACY ON CIFAR-100

we report the per-class retain accuracies before and after unlearning for ResNet18 when forgetting class 0. Table 18 shows the accuracy of all 99 retained classes, together with the accuracy change (after–before). The results indicate that performance changes on most retained classes are small and roughly symmetric around zero, and no clear cluster of semantically similar classes suffers systematic degradation. This is consistent with our adaptive mask mechanism: parameters that are shared between the forget class and semantically related retain classes tend to have large $S_{retain}$, thus failing the ratio-based selection criterion and being automatically protected from perturbation. Consequently, IGNP primarily modifies parameters specific to the forget class, preserving the representation quality of other classes.

## A.12 BASELINE METHODS AND PARAMETER SETTINGS

This section provides detailed parameter settings for all baseline methods compared in our experiments. Tables 20 through 19 summarize these parameters across different unlearning approaches, highlighting the key differences in their implementation strategies.

Table 15: Mean ± Std of accuracy (%) over three runs for the GKT baseline.

| Dataset | Model | Retention Set Result | Forgotten Set Result |
|---|---|---|---|
| MNIST | LeNet | 98.10±0.32 | 0.00±0.00 |
| SVHN | LeNet | 74.43±3.66 | 0.00±0.00 |
| CIFAR-10 | LeNet | 23.39±3.20 | 0.00±0.00 |
| MNIST | ResNet9 | 94.85±0.44 | 0.00±0.00 |
| SVHN | ResNet9 | 87.09±0.52 | 0.00±0.00 |
| CIFAR-10 | ResNet9 | 16.44±0.48 | 0.00±0.00 |
| CIFAR-100 | ResNet9 | 1.82±0.48 | 0.00±0.00 |
| CIFAR-10 | ResNet18 | 19.66±2.14 | 0.00±0.00 |
| CIFAR-100 | ResNet18 | 1.74±0.20 | 0.00±0.00 |

Table 16: Mean ± Std of accuracy (%) over three runs for the SSD baseline.

| Dataset | Model | Retention Set Result | Forgotten Set Result |
|---|---|---|---|
| MNIST | LeNet | 99.35±0.01 | 0.00±0.00 |
| SVHN | LeNet | 89.12±0.10 | 0.00±0.00 |
| CIFAR-10 | LeNet | 72.99±0.39 | 0.00±0.00 |
| MNIST | ResNet9 | 95.30±2.25 | 0.00±0.00 |
| SVHN | ResNet9 | 86.96±0.62 | 0.00±0.00 |
| CIFAR-10 | ResNet9 | 88.69±3.99 | 0.00±0.00 |
| CIFAR-100 | ResNet9 | 71.49±0.31 | 0.00±0.00 |
| CIFAR-10 | ResNet18 | 93.81±0.28 | 0.00±0.00 |
| CIFAR-100 | ResNet18 | 76.38±0.67 | 0.00±0.00 |

## A.13 LIMITATIONS

As a zero-shot, data-free framework, IGNP is designed for class-level unlearning by synthesizing representative class information. This approach inherently limits its scope to entire classes,precluding the more granular task of sample-specific forgetting. In line with current approximate unlearning methods, our validation relies on empirical evidence rather than formal guarantees, highlighting a clear path for future work on certified erasure.

Table 17: Mean ± Std of accuracy (%) over three runs for the UnSIR baseline.

| Dataset | Model | Retention Set Result | Forgotten Set Result |
|---|---|---|---|
| MNIST | LeNet | 97.51±0.24 | 0.00±0.00 |
| SVHN | LeNet | 71.74±3.89 | 0.00±0.00 |
| CIFAR-10 | LeNet | 42.79±1.79 | 0.00±0.00 |
| MNIST | ResNet9 | 96.59±1.26 | 0.00±0.00 |
| SVHN | ResNet9 | 87.96±0.52 | 0.00±0.00 |
| CIFAR-10 | ResNet9 | 65.25±2.07 | 0.00±0.00 |
| CIFAR-100 | ResNet9 | 35.50±2.13 | 0.00±0.00 |
| CIFAR-10 | ResNet18 | 52.11±2.21 | 0.00±0.00 |
| CIFAR-100 | ResNet18 | 35.08±0.98 | 0.00±0.00 |

Table 18: Per-class retain accuracy on CIFAR-100 (ResNet18) before and after unlearning class 0.

| Idx | $Acc_{original}$ | $Acc_{forget}$ | Change | Idx | $Acc_{original}$ | $Acc_{forget}$ | Change | Idx | $Acc_{original}$ | $Acc_{forget}$ | Change |
|---|---|---|---|---|---|---|---|---|---|---|---|
| 1 | 95.0 | 90.0 | -5.0 | 2 | 60.0 | 57.0 | -3.0 | 3 | 66.0 | 63.0 | -3.0 |
| 4 | 68.0 | 65.0 | -3.0 | 5 | 86.0 | 85.0 | -1.0 | 6 | 88.0 | 86.0 | -2.0 |
| 7 | 77.0 | 75.0 | -2.0 | 8 | 89.0 | 90.0 | 1.0 | 9 | 88.0 | 84.0 | -4.0 |
| 10 | 56.0 | 57.0 | 1.0 | 11 | 62.0 | 64.0 | 2.0 | 12 | 83.0 | 83.0 | 0.0 |
| 13 | 78.0 | 81.0 | 3.0 | 14 | 69.0 | 69.0 | 0.0 | 15 | 87.0 | 88.0 | 1.0 |
| 16 | 79.0 | 78.0 | -1.0 | 17 | 89.0 | 89.0 | 0.0 | 18 | 64.0 | 69.0 | 5.0 |
| 19 | 69.0 | 68.0 | -1.0 | 20 | 91.0 | 92.0 | 1.0 | 21 | 91.0 | 91.0 | 0.0 |
| 22 | 80.0 | 78.0 | -2.0 | 23 | 86.0 | 89.0 | 3.0 | 24 | 85.0 | 87.0 | 2.0 |
| 25 | 73.0 | 73.0 | 0.0 | 26 | 80.0 | 72.0 | -8.0 | 27 | 69.0 | 67.0 | -2.0 |
| 28 | 88.0 | 87.0 | -1.0 | 29 | 82.0 | 82.0 | 0.0 | 30 | 66.0 | 73.0 | 7.0 |
| 31 | 79.0 | 82.0 | 3.0 | 32 | 70.0 | 71.0 | 1.0 | 33 | 70.0 | 74.0 | 4.0 |
| 34 | 80.0 | 70.0 | -10.0 | 35 | 59.0 | 52.0 | -7.0 | 36 | 88.0 | 82.0 | -6.0 |
| 37 | 79.0 | 81.0 | 2.0 | 38 | 76.0 | 76.0 | 0.0 | 39 | 93.0 | 92.0 | -1.0 |
| 40 | 70.0 | 67.0 | -3.0 | 41 | 90.0 | 91.0 | 1.0 | 42 | 72.0 | 73.0 | 1.0 |
| 43 | 86.0 | 88.0 | 2.0 | 44 | 53.0 | 51.0 | -2.0 | 45 | 66.0 | 68.0 | 2.0 |
| 46 | 50.0 | 55.0 | 5.0 | 47 | 68.0 | 68.0 | 0.0 | 48 | 96.0 | 96.0 | 0.0 |
| 49 | 90.0 | 90.0 | 0.0 | 50 | 60.0 | 62.0 | 2.0 | 51 | 79.0 | 77.0 | -2.0 |
| 52 | 68.0 | 63.0 | -5.0 | 53 | 89.0 | 83.0 | -6.0 | 54 | 88.0 | 84.0 | -4.0 |
| 55 | 50.0 | 55.0 | 5.0 | 56 | 93.0 | 92.0 | -1.0 | 57 | 77.0 | 77.0 | 0.0 |
| 58 | 89.0 | 88.0 | -1.0 | 59 | 71.0 | 73.0 | 2.0 | 60 | 89.0 | 87.0 | -2.0 |
| 61 | 72.0 | 64.0 | -8.0 | 62 | 75.0 | 79.0 | 4.0 | 63 | 69.0 | 67.0 | -2.0 |
| 64 | 66.0 | 65.0 | -1.0 | 65 | 65.0 | 64.0 | -1.0 | 66 | 85.0 | 84.0 | -1.0 |
| 67 | 68.0 | 66.0 | -2.0 | 68 | 94.0 | 95.0 | 1.0 | 69 | 82.0 | 81.0 | -1.0 |
| 70 | 78.0 | 78.0 | 0.0 | 71 | 84.0 | 83.0 | -1.0 | 72 | 53.0 | 51.0 | -2.0 |
| 73 | 62.0 | 60.0 | -2.0 | 74 | 67.0 | 63.0 | -4.0 | 75 | 92.0 | 94.0 | 2.0 |
| 76 | 91.0 | 92.0 | 1.0 | 77 | 72.0 | 74.0 | 2.0 | 78 | 68.0 | 70.0 | 2.0 |
| 79 | 84.0 | 83.0 | -1.0 | 80 | 65.0 | 66.0 | 1.0 | 81 | 80.0 | 80.0 | 0.0 |
| 82 | 92.0 | 93.0 | 1.0 | 83 | 79.0 | 77.0 | -2.0 | 84 | 71.0 | 70.0 | -1.0 |
| 85 | 91.0 | 91.0 | 0.0 | 86 | 78.0 | 76.0 | -2.0 | 87 | 87.0 | 80.0 | -7.0 |
| 88 | 88.0 | 88.0 | 0.0 | 89 | 86.0 | 91.0 | 5.0 | 90 | 85.0 | 85.0 | 0.0 |
| 91 | 87.0 | 89.0 | 2.0 | 92 | 71.0 | 63.0 | -8.0 | 93 | 67.0 | 68.0 | 1.0 |
| 94 | 96.0 | 93.0 | -3.0 | 95 | 81.0 | 80.0 | -1.0 | 96 | 73.0 | 74.0 | 1.0 |
| 97 | 84.0 | 86.0 | 2.0 | 98 | 61.0 | 67.0 | 6.0 | 99 | 80.0 | 77.0 | -3.0 |

Table 19: Our Method (IGSP) Parameters

| Parameter | Value | Description |
|---|---|---|
| $\lambda$ | 10 | Unlearning regularization strength |
| Coverage Search Range | [0.011, 0.013] for CIFAR-100 [0.04, 0.05] for 10-class datasets | Task-specific coverage bounds |

Table 20: Retrain Method Parameters

| Parameter | Value | Description |
|---|---|---|
| Epochs | 200 | Complete model retraining on retain set |
| Early Stopping | Patience=10 | Based on validation accuracy |
| Other Parameters | Same as Table 1 | Uses original training hyperparameters |

Table 21: Fine-tuning Method Parameters

| Parameter | Value | Description |
| --- | --- | --- |
| Epochs | 20 | Complete model finetuning on retain set |
| Learning Rate | 0.001 | Adam optimizer |
| Training Epochs | 20 | Reduced epochs for efficiency |
| Weight Decay | 5e-4 | Regularization parameter |
| LR Scheduler | CosineAnnealing | $\eta_{\min}$ = initial_lr $\times$ 0.01 |
| ResNet18 Adjustment | Initial LR $\times$ 0.1 | Architecture-specific tuning |

Table 22: SSD Method Parameters

| Parameter | Value | Description |
| --- | --- | --- |
| Selection Weighting ($\alpha$) | 10 (default)
100 for LeNet on CIFAR-10
90 for ResNet-9 on CIFAR-100
90 for ResNet-18 on CIFAR-100 | Parameter importance weighting |
| Dampening Constant ($\lambda$) | 1 | Synaptic dampening strength |

Table 23: GKT Method Parameters

| Parameter | Value | Description |
| --- | --- | --- |
| Iterations | 200 | Knowledge transfer iterations |
| Learning Rate | 0.001 | Optimization learning rate |
| KL Temperature | 1.0 | Knowledge distillation temperature |
| AT Coefficient ($\beta$) | 250.0 | Attention transfer weight |
| Convergence Threshold | 0.01 | Stopping criterion |
| Batch Size | 256 | Training batch size |

Table 24: UNSIR Method Parameters

| Parameter | Value | Description |
| --- | --- | --- |
| Noise Epochs | 5 | Synthetic noise generation |
| Noise Steps | 8 | Steps per noise epoch |
| Noise LR | 0.1 | Noise optimization rate |
| Noise Weight | 0.1 | Noise regularization |
| Impair Batches | 20 | Model impairment batches |
| Impair LR | 0.02 | Impairment learning rate |
| Repair LR | 0.01 | Model repair learning rate |

