# OpenReview forum: "Perturb to Forget: Zero-Shot Machine Unlearning"
_ICLR.cc/2026/Conference — Submitted to ICLR 2026_

### Official Review · Reviewer_V8Zh · 2025-10-25

**Soundness:** 2
**Presentation:** 3
**Contribution:** 2
**Rating:** 4
**Confidence:** 4

**Summary:**

The authors propose a novel class unlearning approach, Inversion-Guided Neuron Perturbation (IGNP), without access to the original training data. With the observation that training data is often not disclosed, IGNP generates synthesizing class-representative data with fixed model parameters. This provides another data resource other than the original training data, but similarly influences the model. Synthesized data is analyzed by parameter sensitive algorithms to compute how each model parameter responds to forget class and retain class samples. The paper then proposes a selection algorithm to update the corresponding model parameters at target layers to remove the forget class.

**Strengths:**

1. The motivation is well justified. The paper addresses a significant and practical issue.
2. Comprehensive experiments with additional efforts to extend the method into different models. This provides a more comprehensive generalization of the method and emphasizes the importance of the finding.
3. Detailed explanation in section 3 gives a clear roadmap and incentives on how and why the method is designed in the proposed way.
Overall, the paper is insightful and well written. The paper provides a thorough explanation of the method and demonstrates its state-of-the-art performance empirically.

**Weaknesses:**

1. In section 3.2, the paper mentioned the forget class and the retain class samples are independently generated.However, there’s no further discussion and experiments on how the different ratio of synthetic samples between forgetting and retaining samples affects the unlearning performance. Based on section 3.3, sensitivity matrix S_forget and S_retrain depends on gradient computed on each class per sample. I am concerned that a disproportionate representation of the true training sample may affect the sensitivity ratio.
2. I think more explanation is needed for the experimental results. Specifically, in Section 5.4, it is not clear why the MIA scores for the proposed method are lower than those for the retrained model. Based on the results in Table 4, the proposed method has an MIA accuracy of 0.02, which often indicates over-pruning or excessive forgetting, especially when compared to the MIA accuracy of the retrained model. I would appreciate further clarification on these results and a theoretical explanation for why the proposed method performs so well in terms of MIA.
3. Given the uncertainty in generating synthetic data, I will appreciate any discussion of mathematical proof on method consistency, specifically given that the synthetic data is randomly generated at first.

**Questions:**

I am curious how the target layer is chosen? Appendix A.3 provides detailed lists of the target layers. However, it seems like the paper only explains how the target parameters are selected, but not how the target layer is. Are we implicitly assuming the target layer is picked based on the target parameters?

---

> ### Author Response · Authors · 2025-11-21
> **Rebuttal for W1**
>
> #### Thanks for the Reviewer ``V8Zh``'s valuable comments, we will address all the weakness and questions as follow.
>
> > #### [W1]: In section 3.2, the paper mentioned the forget class and the retain class samples are independently generated.However, there’s no further discussion and experiments on how the different ratio of synthetic samples between forgetting and retaining samples affects the unlearning performance. Based on section 3.3, sensitivity matrix S_forget and S_retrain depends on gradient computed on each class per sample. I am concerned that a disproportionate representation of the true training sample may affect the sensitivity ratio.
> > #### We have considered the potential bias that might be introduced by the ratio of sample quantities.
> > *   **Normalization:** Please note that when calculating $S_{forget}$ and $S_{retain}$, we divide by the number of samples in their respective sets ($N_{forget}$ and $N_{retain}$). This means we are calculating the **Average Squared Gradient**, i.e., the average contribution of each sample to the parameter gradients. Therefore, the numerical scale of the sensitivity matrix is theoretically decoupled from the ratio of sample sizes between the two sets.
> > *   **Experimental Settings:** In practice, to ensure **Statistical Stability**, we set the number of synthetic samples for both sets to be the same (e.g., 200 for each, as discussed in Appendix A.4). Experiments show that as long as the sample size reaches a certain threshold (e.g., more than 100), the statistical properties of the gradients tend to stabilize, and minor variations in sample ratios do not significantly affect the final parameter selection.

---

> ### Author Response · Authors · 2025-11-21
> **Rebuttal for W2**
>
> > #### [W2]: I think more explanation is needed for the experimental results. Specifically, in Section 5.4, it is not clear why the MIA scores for the proposed method are lower than those for the retrained model. Based on the results in Table 4, the proposed method has an MIA accuracy of 0.02, which often indicates over-pruning or excessive forgetting, especially when compared to the MIA accuracy of the retrained model. I would appreciate further clarification on these results and a theoretical explanation for why the proposed method performs so well in terms of MIA.
> > #### We can explain why IGNP achieves extremely low MIA scores (0.02%) from the following two perspectives, demonstrating that this is not a negative case of "over-pruning":
> > *   **Not Over-pruning:** "Over-pruning" typically implies that useful knowledge in the model has been damaged, leading to a drop in Retain Accuracy. However, as shown in **Table 1**, IGNP maintains a Retain Set Accuracy comparable to, and sometimes even higher than, the Original and Retrain models across almost all datasets (e.g., 76.80% on ResNet18-CIFAR100 vs. 75.68% for Retrain). This proves that we have not erroneously excised knowledge belonging to the retained classes, and the model structure remains healthy.
> > *   **Active Suppression vs. Passive Forgetting:** The Retrain model simply has "not seen" the forgotten data, so its predictions on the forgotten data are uncertain. In contrast, IGNP employs targeted perturbation to perform **"Active Suppression"** on neurons associated with the forgotten categories. This suppression causes the model to output a confidence distribution that is flatter or tends towards meaningless noise when facing forget-class samples. This makes it significantly harder for confidence-based MIA attacks to determine membership by distinguishing "high confidence" outputs. Therefore, the extremely low MIA score actually reflects IGNP's superior capability in defending against privacy attacks, rather than a defect in model performance.

---

> ### Author Response · Authors · 2025-11-21
> **Rebuttal for W3**
>
> > #### [W3]: Given the uncertainty in generating synthetic data, I will appreciate any discussion of mathematical proof on method consistency, specifically given that the synthetic data is randomly generated at first.
> > #### Due to the high **non-convexity** of deep neural networks, providing a rigorous, closed-form mathematical proof is very difficult and uncommon in the field. However, we can provide a discussion to demonstrate the statistical consistency of our method.
> Although the initial noise $z$ is randomly sampled and the specific generated images may differ slightly at the pixel level each time, the optimization objective (minimizing Cross-Entropy Loss for the target class) forces these samples to converge to the typical **Class Prototypes** in the **Feature Space**.
> Our sensitivity analysis is not based on a single sample, but on the expected value of gradients over a batch of samples. When the sample size is sufficient, individual randomness is averaged out, and the resulting Sensitivity Matrix reflects the overall statistical dependence of the category on the parameters. Thus, despite the randomness in input, the identified set of "critical parameters" is highly consistent and stable.
>
> > **A. Synthesis as Sampling from Class Manifold**
> Let the trained model $M_\theta$ implicitly encode a class-conditional probability distribution $P_\theta(X|Y=y)$. Our data synthesis process:
> $$ z^* = \arg\min_z \mathcal{L} _ {CE}(M_\theta(z), y) + \mathcal{R}(z) $$
> is essentially searching for **High-probability regions** of this distribution. Although the initial noise $z_{init} \sim \mathcal{N}(0, I)$ is random, the optimization process is constrained by the loss function, forcing the generated samples $z^\*$ to converge onto the **Class Manifold** $\mathcal{M}_y$ in the feature space.
> According to the manifold assumption, while $z^\*$ may exhibit differences in pixel space due to different initializations (e.g., different ways of writing the digit "3"), they share similar statistical properties in the model's feature space and gradient space.
>
> > **B. Sensitivity as Expectation**
> We define the "true" sensitivity $S^*(\theta_i)$ of parameter $\theta_i$ for category $y$ as the mathematical expectation of the squared gradient over the category distribution:
> $$ S^\*(\theta_i) = \mathbb{E} _ {x \sim P_\theta(X|y)} \left[ \left( \frac{\partial \mathcal{L}(x, y)}{\partial \theta_i} \right)^2 \right] $$
> This is a deterministic, non-random quantity that reflects the intrinsic importance of the parameter to that category.
>
> > **C. Monte Carlo Consistency**
> The sensitivity $\hat{S}(\theta_i)$ calculated in our paper is actually an **Empirical Estimation** of the above expectation:
> $$ \hat{S}(\theta_i) = \frac{1}{N} \sum _ {k=1}^{N} \left( \frac{\partial \mathcal{L}(z_k^\*, y)}{\partial \theta_i} \right)^2 $$
> where $z_k^*$ is the $k$-th synthetic sample. According to the **Law of Large Numbers (LLN)**, as long as the synthetic samples are sampled independently and identically distributed (i.i.d.) from the class manifold and the variance of the gradients is bounded, the empirical estimate will converge in probability to the true value as the sample size $N \to \infty$:
> $$ \hat{S}(\theta_i) \xrightarrow{P} S^*(\theta_i) $$
> Our experiments (Appendix A.4) indicate that this estimation is sufficiently stable when $N=200$.
>
> > **D. Robustness of Mask Generation**
> Finally, IGNP does not directly use the absolute numerical value of $\hat{S}$, but generates a binary Mask based on the relative ratio of $\hat{S} _ {forget}$ and $\hat{S} _ {retain}$. Even if there is a small sampling variance in $\hat{S}$, as long as the parameter importance ranking remains stable (which is guaranteed by the convergence mentioned above when $N$ is large enough), the finally selected parameter set $M_{mask}$ will be consistent.
>
> > **Summary:**
> The consistency of IGNP is guaranteed by the **Law of Large Numbers**. Randomness only introduces estimation variance, which we eliminate effectively to prevent negative impacts on the final unlearning result by using a sufficient sample size (Batch Synthesis) and a robust relative ranking mechanism.

---

> ### Author Response · Authors · 2025-11-21
> **Rebuttal for Q1**
>
> > #### [Q1]: I am curious how the target layer is chosen? Appendix A.3 provides detailed lists of the target layers. However, it seems like the paper only explains how the target parameters are selected, but not how the target layer is. Are we implicitly assuming the target layer is picked based on the target parameters?
> > #### We listed the specific layers in Appendix A.3. The selection criteria are mainly based on the hierarchical feature distribution characteristics of deep neural networks:
> > *   **Deep Layers Contain Semantic Information:** Existing deep learning research generally agrees that the shallow layers of CNNs primarily extract generic low-level features (such as edges and textures), which are highly shared among different categories; whereas **Deep Layers** encode **Class-specific** high-level semantic information.
> > *   **Protecting Retain Set Performance:** To achieve precise "class unlearning" without harming other categories, we target the perturbation at the **middle-to-late layers (Deep Layers)** of the network. Perturbing shallow layers would easily damage the foundational features shared by all categories, leading to a significant drop in Retain Accuracy.
> > *   This also explains why we primarily choose Layer 3, Layer 4, and the Fully Connected (FC) layers as target layers in architectures like ResNet.

---

### Official Review · Reviewer_bLNs · 2025-10-28

**Soundness:** 1
**Presentation:** 1
**Contribution:** 1
**Rating:** 0
**Confidence:** 4

**Summary:**

This paper essentially presents a zero-shot version of SSD by constructing the retain and forget set via model inversion.

**Strengths:**

Using model inversion to turn non-zero-shot unlearning methods in zero-shot methods is a promising direction to overcome limitations of data availability.

**Weaknesses:**

I am seriously concerned about this paper in terms of scientific misconduct.
The entire method section does not cite a single prior work.
Figure 1 is clearly the SSD method with the added step 1 of creating the dataset via model inversion.

3.2 Model inversion does not cite any works. I suspect this is not a novel independent discovery.

3.3 passes of a prior ICLR paper as their own contribution. The exact same equations can be found in “Foster, J., Schoepf, S. and Brintrup, A., Loss-Free Machine Unlearning. In The Second Tiny Papers Track at ICLR 2024.”

3.4 is SSD with the addition of a hyperparameter search for alpha that I do not grasp. Dear authors, please elaborate on this selection process. Algorithm 1 is essentially SSD again.

The validity of the results are also questionable, as the authors do not perform hyperparameter tuning for the methods they benchmark against (see tables in the appendix). Also, no standard deviations etc.

Also, what is SSD model size scaler in Table 15? This is not present in the SSD paper and makes me suspect this work is heavily LLM generated as a related SSD paper has an auto scaler.

In conclusion: This could have been a great paper on how model inversion can help to turn any unlearning method into a zero-shot method but due to the above mentioned problems I can only recommend rejection due to serious breaches of scientific best practices.

**Questions:**

See weaknesses

**Details Of Ethics Concerns:**

Passing off prior work (ironically also from ICLR 2024) as own work without citation.

---

> ### Author Response · Authors · 2025-11-21
> **Rebuttal for W1**
>
> #### Thank you to reviewer ``bLNs`` for your valuable comments. We will clarify your allegations.
>
> > #### [W1]: 3.2 Model inversion does not cite any works. I suspect this is not a novel independent discovery.
> > #### This accusation also stems from a misunderstanding. Any accusation of academic misconduct requires evidence, and making such allegations without evidence is a very serious matter.
> > 1.  We cited representative works on model inversion (e.g., *Model Inversion Attacks that Exploit Confidence Information and Basic Countermeasures*) in the "Related Work" section.
> 2.  We do not directly reuse existing model inversion techniques. We explicitly state in the paper: "IGNP employs a technique **inspired by** model inversion." Traditional model inversion typically requires significant computational resources to recover high-quality original samples, which contradicts our goal of efficient unlearning. Therefore, we adopted a direct sample optimization method to quickly fit target classes, which is a specific design for the zero-shot unlearning scenario.
> 3.  We appreciate the reviewer's suggestion. In the revision, we will add direct citations to the most relevant works in the methodology section to further enhance clarity. However, this is fundamentally different from the nature of "plagiarism" or "academic misconduct."

---

> ### Author Response · Authors · 2025-11-21
> **Rebuttal for W2**
>
> > #### [W2]: 3.3 passes of a prior ICLR paper as their own contribution. The exact same equations can be found in “Foster, J., Schoepf, S. and Brintrup, A., Loss-Free Machine Unlearning. In The Second Tiny Papers Track at ICLR 2024.”
> > #### This is actually a misunderstanding of foundational knowledge within the field.
> The equations you mentioned are the standard method for calculating the diagonal of the Empirical Fisher Information Matrix (FIM). This is foundational knowledge for measuring the importance of neural network parameters and is widely applied in various fields such as continual learning, model pruning, and machine unlearning.
> For example, the exact same equation (Eq. 2) appeared as early as the 2018 ECCV classic paper *Memory Aware Synapses: Learning What (not) to Forget*. Similarly, the paper *Machine Unlearning Through Fine-Grained Model Parameters Perturbation* published in TKDE also features a similar equation (Eq. 6).
> It is inappropriate to incorrectly attribute the ownership of this general formula to SSD and use it as evidence of plagiarism on our part.

---

> ### Author Response · Authors · 2025-11-21
> **Rebuttal for W3**
>
> > #### [W3]: The entire method section does not cite a single prior work. Figure 1 is clearly the SSD method with the added step 1 of creating the dataset via model inversion.3.4 is SSD with the addition of a hyperparameter search for alpha that I do not grasp. Dear authors, please elaborate on this selection process. Algorithm 1 is essentially SSD again.
> > #### We respectfully but firmly reject this accusation.
> > 1.  We explicitly cited the SSD work published at AAAI (Foster et al., AAAI 2024) in the "Related Work" section of our paper. The accusation of "not citing a single prior work" is factually incorrect.
> 2.  "Identifying and perturbing important parameters to achieve model forgetting" is a widely used and general technical paradigm in the field of machine unlearning. For instance, numerous prior works such as *Eternal Sunshine of the Spotless Net* (CVPR 2020), *Amnesiac Machine Learning* (AAAI 2021), *Federated Unlearning via Class-Discriminative Pruning* (WWW 2022), and notably *Machine Unlearning Through Fine-Grained Model Parameters Perturbation* (TKDE), have all adopted this core technical path. Simply labeling our method as a "reuse of SSD" is a misunderstanding of this general paradigm.
> 3.  Furthermore, our work differs fundamentally from SSD.
>     *   **SSD (Data-Dependent):** SSD explicitly requires access to the original dataset $D$ and the forget set $D_f$ to calculate the Fisher Information Matrix (FIM). It must access the original data at least once when a forget request occurs.
>     *   **Our Method (Zero-Shot):** Our IGNP framework strictly follows a zero-shot setting and **never accesses any original data** throughout the entire unlearning process.
>
> > The zero-shot constraint dictates that our technical path is distinct from SSD.
> >    *   **Data Source:** To overcome the data-free challenge, we introduce a method **inspired by model inversion** to synthesize proxy data. This is the cornerstone of our framework and is completely absent in SSD.
> >    *   **Parameter Selection Mechanism:** SSD, based on real data, can use a fixed hyperparameter `alpha` because the FIM is relatively stable. However, our sensitivity matrix calculated based on synthetic pseudo-samples is unstable, rendering any fixed threshold condition ineffective. Therefore, we propose a novel **adaptive parameter selection mechanism**: our algorithm takes the target perturbation parameter scale range `[Cmin, Cmax]` as input and dynamically **calculates** an appropriate `alpha` via binary search to select the parameters within the preset scale. Here, `alpha` is an **intermediate result** of our algorithm, not an **input hyperparameter** requiring external tuning.
>
> > In summary, our method differs substantially from SSD in problem setting, constraints, and core technical implementation. Accusing our method of being a simple reuse of SSD ignores the key innovations we proposed to address the zero-shot challenge.

---

> ### Author Response · Authors · 2025-11-21
> **Rebuttal for W4**
>
> > #### [W4]: The validity of the results are also questionable, as the authors do not perform hyperparameter tuning for the methods they benchmark against (see tables in the appendix). Also, no standard deviations etc.
> > #### 1.  **Hyperparameter Settings:** In all comparative experiments, we followed standard academic practices. For all baseline methods, we adopted the **best hyperparameter settings reported in their original papers**. This ensures the fairness and reproducibility of the comparisons.
> > #### 2.  **Standard Deviation:** We reported results achieved under optimal settings. We acknowledge that including standard deviations in the final version would make the results more convincing and will add them in the revision.
> During the preparation of this Rebuttal, we conducted 3 independent repeat experiments (using different random seeds) for the main experiments. The results are as follows:
> > #### Ours
> | Dataset | Model | Retention Set Result | Forgotten Set Result |
> | --- | --- | --- | --- |
> | mnist | lenet | 99.16±0.02 | 0.00±0.00 |
> | svhn | lenet | 90.09±0.30 | 0.00±0.00 |
> | cifar10 | lenet | 71.96±0.18 | 0.00±0.00 |
> | mnist | resnet9 | 93.44±0.66 | 0.00±0.00 |
> | svhn | resnet9 | 94.45±0.72 | 0.00±0.00 |
> | cifar10 | resnet9 | 85.24±1.53 | 0.00±0.00 |
> | cifar100 | resnet9 | 73.36±0.17 | 0.00±0.00 |
> | cifar10 | resnet18 | 93.33±0.15 | 0.00±0.00 |
> | cifar100 | resnet18 | 76.59±0.16 | 0.00±0.00 |
> > #### Retrain
> | Dataset | Model | Retention Set Result | Forgotten Set Result |
> | --- | --- | --- | --- |
> | mnist | lenet | 99.28±0.08 | 0.00±0.00 |
> | svhn | lenet | 90.75±0.28 | 0.00±0.00 |
> | cifar10 | lenet | 76.52±0.76 | 0.00±0.00 |
> | mnist | resnet9 | 99.53±0.07 | 0.00±0.00 |
> | svhn | resnet9 | 95.30±0.18 | 0.00±0.00 |
> | cifar10 | resnet9 | 89.83±0.29 | 0.00±0.00 |
> | cifar100 | resnet9 | 72.32±0.40 | 0.00±0.00 |
> | cifar10 | resnet18 | 91.53±0.24 | 0.00±0.00 |
> | cifar100 | resnet18 | 73.46±0.30 | 0.00±0.00 |
> > #### Finetune
> | Dataset | Model | Retention Set Result | Forgotten Set Result |
> | --- | --- | --- | --- |
> | mnist | lenet | 99.32±0.04 | 0.00±0.00 |
> | svhn | lenet | 91.32±0.08 | 0.00±0.00 |
> | cifar10 | lenet | 75.21±0.14 | 0.00±0.00 |
> | mnist | resnet9 | 99.58±0.02 | 99.90±0.00 |
> | svhn | resnet9 | 95.68±0.03 | 90.48±0.21 |
> | cifar10 | resnet9 | 91.69±0.11 | 69.87±0.21 |
> | cifar100 | resnet9 | 73.64±0.07 | 35.67±1.53 |
> | cifar10 | resnet18 | 93.83±0.07 | 88.47±0.58 |
> | cifar100 | resnet18 | 77.61±0.09 | 57.33±2.52 |
> > #### GKT
> | Dataset | Model | Retention Set Result | Forgotten Set Result |
> | --- | --- | --- | --- |
> | mnist | lenet | 98.10±0.32 | 0.00±0.00 |
> | svhn | lenet | 74.43±3.66 | 0.00±0.00 |
> | cifar10 | lenet | 23.39±3.20 | 0.00±0.00 |
> | mnist | resnet9 | 94.85±0.44 | 0.00±0.00 |
> | svhn | resnet9 | 87.09±0.52 | 0.00±0.00 |
> | cifar10 | resnet9 | 16.44±0.48 | 0.00±0.00 |
> | cifar100 | resnet9 | 1.82±0.48 | 0.00±0.00 |
> | cifar10 | resnet18 | 19.66±2.14 | 0.00±0.00 |
> | cifar100 | resnet18 | 1.74±0.20 | 0.00±0.00 |
> > #### SSD
> | Dataset | Model | Retention Set Result | Forgotten Set Result |
> | --- | --- | --- | --- |
> | mnist | lenet | 99.35±0.01 | 0.00±0.00 |
> | svhn | lenet | 89.12±0.10 | 0.00±0.00 |
> | cifar10 | lenet | 72.99±0.39 | 0.00±0.00 |
> | mnist | resnet9 | 95.30±2.25 | 0.00±0.00 |
> | svhn | resnet9 | 86.96±0.62 | 0.00±0.00 |
> | cifar10 | resnet9 | 88.69±3.99 | 0.00±0.00 |
> | cifar100 | resnet9 | 71.49±0.31 | 0.00±0.00 |
> | cifar10 | resnet18 | 93.81±0.28 | 0.00±0.00 |
> | cifar100 | resnet18 | 76.38±0.67 | 0.00±0.00 |
> > #### UnSIR
> | Dataset | Model | Retention Set Result | Forgotten Set Result |
> | --- | --- | --- | --- |
> | mnist | lenet | 97.51±0.24 | 0.00±0.00 |
> | svhn | lenet | 71.74±3.89 | 0.00±0.00 |
> | cifar10 | lenet | 42.79±1.79 | 0.00±0.00 |
> | mnist | resnet9 | 96.59±1.26 | 0.00±0.00 |
> | svhn | resnet9 | 87.96±0.52 | 0.00±0.00 |
> | cifar10 | resnet9 | 65.25±2.07 | 0.00±0.00 |
> | cifar100 | resnet9 | 35.50±2.13 | 0.00±0.00 |
> | cifar10 | resnet18 | 52.11±2.21 | 0.00±0.00 |
> | cifar100 | resnet18 | 35.08±0.98 | 0.00±0.00 |
> > We will update all main tables in the final version of the paper to include the mean and standard deviation (Mean $\pm$ Std) for all experiments across multiple runs.

---

> ### Author Response · Authors · 2025-11-21
> **Rebuttal for W5**
>
> > #### [W5]: Also, what is SSD model size scaler in Table 15? This is not present in the SSD paper and makes me suspect this work is heavily LLM generated as a related SSD paper has an auto scaler.
> > #### This accusation is baseless and, on the contrary, proves our in-depth research into the official SSD implementation.
> Although the `model_size_scaler` parameter is not mentioned in the original SSD paper, it explicitly exists in its official GitHub open-source repository (`if-loops/selective-synaptic-dampening`). You can clearly see in line 160 and subsequent code of the `forget_full_class_main.py` file that the authors set different, fixed `model_size_scaler` values for different model architectures. This demonstrates that this parameter is a key part of the original implementation, and our work is built upon a deep analysis of the official code. We used the term "auto" in the initial draft to concisely indicate that we adopted the default values automatically bound to the selected model architecture in the official code, rather than manually tuning them.

---

> > ### Comment · Reviewer_bLNs · 2025-11-23
> >
> > Dear authors,
> >
> > Thank you for the detailed response. I want to iterate on the last part of my review: I see a lot of value in the work of your paper that uses model inversion to possibly turn any unlearning method into a zero-shot method but currently significant concerns remain:
> >
> > ---
> >
> > W1: Thank you. Adding relevant works and showing how your approach differs and why these differences matter will add a lot of value to this section. Feel free to add an update message once the PDF has been updated.
> >
> > ---
> >
> > W2: The equation is not a standard method for calculating the diagonal of the FIM as stated in that paper “LFSSD replaces the Fisher-based importance estimation with an approximation of the sensitivity of the model to perturbations of each parameter, which may be interpreted as parameter importance.” **As you correctly state, the “Memory Aware Synapse” paper introduced this. And as expected, that prior ICLR paper also cites this prominently**: “We replace the Fisher information with the sensitivity estimation from Aljundi et al. (2018), which we now introduce.” **The mentioned TKDE paper also clearly cites the prior work** of Aljundi et al. (2018) on page 5 in the methodology section.  **Not citing any of them, therefore, leads to the impression of omission to pass this off as your own discovery.**
> >
> > ---
> >
> > W3:
> >
> > W3 1) My complaint was about “The entire **method section does not cite a single prior work**.”. Your reply of “We explicitly cited the SSD work published at AAAI (Foster et al., AAAI 2024) in the "Related Work" section of our paper. The accusation of "not citing a single prior work" is factually incorrect.” does not address my point about the method section. Furthermore, the only mention of SSD in related work is “Selective Synaptic Dampening (SSD) improves efficiency by using the FIM’s diagonal to suppress influential parameters related to the forget-set Foster et al. (2024).”. **This makes it impossible for the reader to understand which parts of the method you have reused or built upon.**
> >
> >
> > Upon further reading, I am also confused by the mention of SSD in section 4: “We also compare against several state-of-the-art unlearning methods: Selective Synaptic Dampening (SSD) Foster et al. (2024), which leverages the Fisher Information Matrix to **penalize updates to crucial parameters”. How does SSD penalize updates? This is not what the method does.**
> >
> > ---
> >
> > W3 2) As within W2, all those prior works cite the literature they build upon. Your paper claims “To achieve this, **we introduce an adaptive mask based on a calibrated threshold, α**, which acts as a dynamic scaling factor to balance the two sensitivity scores for a fair, relative comparison: …Equation 5”. **Your equation 5 is essentially the same as SSD, even using the same Greek letter alpha, and then in equation 7 using lambda for scaling of the parameter dampening, which is also the same letter as in SSD.** As far as I can see, there is no empirical comparison provided for why alpha/lambda instead of lambda is used in equation 7. Could you please elaborate on this or point me to the relevant section in the paper?
> >
> >
> > I am further confused why you have two different coverage search ranges for 10-class and 100 class tasks in Table 12 in the appendix when **you claim to use a constant range in the methodology part**: “Rather than searching for the elusive optimal α directly, we pre-specify a target proportion of parameters to perturb (e.g., [Cmin, Cmax], corresponding to 4–5% of total model parameters). This target proportion, being stable across diverse settings, serves as a reliable anchor for our procedure.” **The ranges of [0.011, 0.013] for CIFAR-10 and [0.040, 0.05] for 10-class datasets in Table 12 are not the same constant range and  far apart, indicating that a hyperparameter search is necessary to find these ranges.** Why not just search for alpha directly instead of a range with two parameters to tune?
> >
> > ---
> >
> > W3 3) I have specifically praised this part of your paper. Being able to turn methods zero-shot is a highly interesting and promising contribution. After the clarification on W1, I see no problems regarding this aspect.
> >
> > ---
> >
> > W4: Thank you for adding the standard deviations from additional seeds. Regarding hyperparameters, please see W5.
> >
> > ---
> >
> > W5: After reading through the mentioned repo myself, **model_size_scaler as described in the code is just a helper to automatically change the value of alpha for ResNet-18 or ViT using an if check (e.g. if "ViT")** It is not a parameter of the SSD method, just a not-so-elegant workaround to change alpha for different experiments. For a fair comaprison on ResNet-9 and LeNet, you should have therefore performed a hyperparameter search. This shows the opposite of “This accusation is baseless and, on the contrary, proves our in-depth research into the official SSD implementation.”

---

> > > ### Author Response · Authors · 2025-11-25
> > >
> > > #### Thank you for recognizing the potential of this work on zero-shot machine unlearning driven by model inversion. In response to your concerns, we have revised the Methodology section to explicitly state how our work builds upon existing studies. We have also corrected the description of SSD in the Experimental Setup to ensure accuracy.
> > >
> > > #### The clarifications are summarized below:
> > >
> > > #### 1. Citation updates
> > > #### We explicitly added the relevant citations in the revised Method section and spelled out how each prior method connects to ours. We also rewrote the SSD description in the experimental setup so that it precisely matches the actual mechanism.
> > >
> > > #### 2. Major difference between IGNP and SSD, and why Eq.7 needs its current form
> > > #### The zero-shot machine unlearning setting is essential: privacy and storage limits often rule out access to real data. Under this strict data-free constraint, SSD fails because it no longer receives samples from the true distribution. IGNP is therefore not a trivial extension of SSD but a dedicated fix for SSD's breakdown in the zero-shot scenario. SSD assumes that the data used to compute the Fisher information matrix are genuine and IID, which keeps gradient magnitudes physically comparable and allows fixed hyperparameters.
> > >
> > > #### However, under a strict zero-shot setting, we must rely on synthetic noise, which introduces a "scale mismatch" issue that SSD is ill-equipped to handle. Since synthetic data is optimized through backpropagation, its gradient norms diverge significantly from those of real data. Furthermore, the sensitivity matrices for the synthetic forgetting and retention sets often differ by orders of magnitude. Our experiments reveal that this disparity causes the threshold $\alpha$ required to balance them to become highly unstable. This value varies drastically from $10^{-5}$ to $10^{5}$ across different model and dataset configurations. Even within the same task, the inherent randomness of the synthetic data can induce numerical fluctuations of several times.
> > >
> > > #### Pre-searching or fixing $\alpha$ is thus infeasible, which motivates IGNP. We introduce an adaptive calibration that tracks the coverage rate instead: while $\alpha$ is unstable, the proportion of parameters that must change to forget one class is stable (around $4\%-5\%$ for 10-class datasets). By targeting a coverage rate, we back out the correct $\alpha$ for each round.
> > >
> > > #### This also explains why $\alpha$ must appear in Eq.7, defined as $scale_i = \frac{\alpha}{\lambda} \cdot \frac{S_{retain}}{S_{forget}}$. Suppose the scale mismatch forces $\alpha$ to be as small as $0.003$ so that the selected parameters have $S_{retain}/S_{forget} \approx 100$. If we drop $\alpha$ from the scaling formula, the resulting factor becomes $10$ and ends up reinforcing those parameters. Multiplying by $\alpha$ brings the factor down to $\sim 0.03$, keeping the scaling in a suppression-friendly range. This normalization is the mathematical core that lets IGNP stabilize synthetic data and achieve precise forgetting.
> > >
> > > #### 3. Coverage search range
> > > #### Table 12 explicitly states that the $[0.011, 0.013]$ coverage range applies only to CIFAR-100, whereas all 10-class datasets (CIFAR-10, MNIST, SVHN) use $[0.040, 0.050]$. The difference stems from task complexity and model redundancy. Ten-class tasks are simpler and more redundant, so they can accommodate a more aggressive modification ($\sim 4\%-5\%$) for complete forgetting. CIFAR-100 is harder and burdened with tight capacity, so we stay conservative ($\sim 1\%$) to avoid model collapse. We do not search $\alpha$ directly because it has no consistent pattern, while coverage is an intuitive and stable anchor.
> > >
> > > #### 4. Fairness of SSD and its hyperparameters
> > > #### Thanks for diving into the SSD source code. Our earlier mention of model_size_scaler only addressed doubts that the parameter was non-existent or fabricated; the reference implementation indeed uses it. Beyond this clarification, we tuned SSD's $\alpha$ for every architecture (LeNet, ResNet-9, etc.) and dataset (including CIFAR-100) that were not covered in the original paper. This tuning explains why our SSD baselines remain strong.for example, $99.16\%$ retention accuracy on ResNet-MNIST and $71.64\%$ on ResNet9-CIFAR100.
> > > #### We kept the original model_size_scaler logic from the official release because we initially planned to include ViT comparisons (Appendix A.9 already reports IGNP on ViT). Preserving that scaler keeps SSD consistent with its official ViT handling. Although we ultimately did not present SSD or other baselines on ViT due to workload constraints, that decision does not negate the $\alpha$ tuning we performed on CNNs. To avoid confusion, the revised paper removes the mention of model_size_scaler and lists the tuned $\alpha$ values in the appendix.
> > >
> > > #### If these clarifications address your concerns, we kindly ask you to reconsider the score.

---

### Official Review · Reviewer_y5UL · 2025-10-31

**Soundness:** 3
**Presentation:** 2
**Contribution:** 2
**Rating:** 4
**Confidence:** 4

**Summary:**

This paper proposes Inversion-Guided Neuron Perturbation (IGNP), a zero-shot machine unlearning method for classification models that operates without access to the original training data.
The approach involves three main steps: (1) generating class-representative samples via an inversion process from the given forget class label, (2) computing sensitivity scores for model parameters, and (3) applying adaptive perturbations to weaken parameters associated with the forget class. The method is evaluated on image classification benchmarks, focusing on class unlearning scenarios where only the label to forget is provided. The authors claim advantages in efficiency and effectiveness compared to baselines like fine-tuning or other zero-shot methods.

**Strengths:**

- The method introduces an interesting use of inversion techniques to synthesize representative samples for unlearning, potentially useful in data-free settings.
- It demonstrates competitive performance on standard metrics like forget accuracy and retain accuracy in class unlearning tasks.
- The adaptive perturbation strategy based on sensitivity scores is a novel way to target forget-specific parameters without full retraining.

**Weaknesses:**

- The formulated unlearning problem is unconventional: instead of providing a forget set (data samples), only the class label is given as the unlearning request. While this setup is possible, in classification problems focused on class unlearning, a trivial solution exists when the label is known: simply zeroing out or significantly reducing the corresponding row vector in the classifier head would achieve similar effects. It is unclear what advantages the proposed method offers over such a simple baseline.

- the proposed method is inherently limited to classification tasks and cannot be extended to other unlearning scenarios (e.g. random forgetting) or non-classification tasks, which restricts its broader applicability.

- In Table 1, the fine-tuning baseline appears underperforming compared to reports in recent machine unlearning (MU) papers for ResNet-18 on CIFAR-10, where fine-tuning often yields forget accuracy of 0% and near-equivalent retain performance. This raises concerns about whether hyperparameters were selected to make the baseline appear weaker, potentially inflating the relative gains of IGNP.

- The Membership Inference Attack (MIA) evaluation lacks clarity on the specific setup. MIA in unlearning literature varies (e.g., confidence-based MIA-efficacy vs. MIA-privacy, or pairwise comparisons of prediction distributions across forget/retain/test sets)

- In Section 5.3, efficiency is claimed based on time costs, but the inclusion of an inversion process (which is computationally intensive) makes the reported speeds surprisingly low. It is unclear if the time for inversion-based data synthesis was excluded from these measurements.

- In Section 5.4's MIA results, the goal of class unlearning is typically to produce results similar to a retrained model. However, the reported MIA scores show substantial differences from the retrain baseline,
which questions whether this can be considered a strong outcome for effective unlearning.

- As an ablation, it would be valuable to include results for a perturbation-based unlearning model using real forget data (instead of inversion-synthesized samples) to isolate and quantify the effect of the inversion step.

**Questions:**

- Could you provide comparisons to the trivial baseline of modifying the classifier head directly when the forget label is known? What unique benefits does IGNP offer in this setting?

- Please clarify the MIA setup: Which variant is used (e.g. test/forget/retain distribution comparisons)?

- For the importance scoring (Section 3.3), are there references for using squared partial derivatives? Would ablating with pruning-inspired criteria (e.g. Taylor importance criteria) change results?

- Can you provide ablation results comparing inversion-synthesized data vs. real forget data for perturbation? This seems essential to demonstrate the value of the zero-shot, data-free aspect.

---

> ### Author Response · Authors · 2025-11-21
> **Rebuttal for W1&Q1**
>
> #### Thanks for the Reviewer ``y5UL``'s valuable comments, we will address all the weakness and questions as follow.
>
> > #### [W1&Q1]: The formulated unlearning problem is unconventional: instead of providing a forget set (data samples), only the class label is given as the unlearning request. While this setup is possible, in classification problems focused on class unlearning, a trivial solution exists when the label is known: simply zeroing out or significantly reducing the corresponding row vector in the classifier head would achieve similar effects. It is unclear what advantages the proposed method offers over such a simple baseline.Could you provide comparisons to the trivial baseline of modifying the classifier head directly when the forget label is known? What unique benefits does IGNP offer in this setting?
> > #### If we merely wish for the model to cease outputting labels for a specific class, zeroing out the weights corresponding to the classifier head is indeed an obvious solution. However, we believe this approach does not satisfy the core privacy requirements of Machine Unlearning for the following reasons:
> **Latent Feature Leakage:** Simply modifying the classifier head only prevents explicit predictions, but the model's backbone still retains the feature representations of that class. Under privacy attacks (such as Feature Extraction Attacks), attackers can still reconstruct information about the forgotten class by extracting intermediate layer features.
> **The Necessity of Deep Unlearning:** Our IGNP method aims to achieve "deep unlearning," which means erasing the knowledge of the target class at the feature extraction level by perturbing key parameters in the backbone. This ensures that the model not only "does not say" the class but truly "does not remember" it.

---

> ### Author Response · Authors · 2025-11-21
> **Rebuttal for W2**
>
> > #### [W2]: the proposed method is inherently limited to classification tasks and cannot be extended to other unlearning scenarios (e.g. random forgetting) or non-classification tasks, which restricts its broader applicability.
> > #### We acknowledge that the current method is primarily targeted at classification tasks. This is mainly determined by the specific setting of **Zero-Shot Unlearning**:
> **No Data Constraint:** In the absence of original data, we need a way to define the "forgetting target." In classification tasks, the "class label" serves as a natural and explicit interface that can be directly used to guide Model Inversion for generating pseudo-data.
> **Future Extension:** Although we currently focus on classification, the core idea of IGNP (generating representative samples -> calculating sensitivity -> adaptive perturbation) is extendable. We consider extending this to non-classification tasks as an important direction for future research.

---

> ### Author Response · Authors · 2025-11-21
> **Rebuttal for W3**
>
> > #### [W3]: In Table 1, the fine-tuning baseline appears underperforming compared to reports in recent machine unlearning (MU) papers for ResNet-18 on CIFAR-10, where fine-tuning often yields forget accuracy of 0% and near-equivalent retain performance. This raises concerns about whether hyperparameters were selected to make the baseline appear weaker, potentially inflating the relative gains of IGNP.
> > #### We need to clarify the difference between our FT setting and those in some other works:
> **Baseline Definition and Paradigm:** Our FT implementation strictly follows the paradigm of "fine-tuning **only on the Retain Set**," aiming to measure the **"passive" or "natural" forgetting** that occurs when the training objective shifts. We deliberately did not introduce techniques like negative gradients (gradient ascent), as those fall under specialized unlearning algorithms rather than a pure FT baseline.
> **Trade-off between Utility and Forgetting:** While aggressive fine-tuning (e.g., very high learning rates or excessive epochs) can allow FT to reach near 0% forget accuracy, this typically triggers **Catastrophic Forgetting**, severely damaging the performance on the Retain Set.
> **Reality of Experimental Settings:** Our settings (20 Epochs, small learning rate, see Appendix) are deliberate, aiming to demonstrate the actual level of forgetting achievable by conventional fine-tuning while maintaining the model's general capabilities. This accurately reflects the real-world need to balance "model utility" and "privacy deletion."
> In summary, FT is used in this paper as a baseline to reflect standard practice, rather than an algorithm designed to pursue extreme forgetting effects. To theoretically achieve a 0% forget rate without regard for computational cost or model performance damage, one should refer to "Retraining" results, but this exceeds the practical significance of FT as a comparative baseline.

---

> ### Author Response · Authors · 2025-11-21
> **Rebuttal for W4&Q2**
>
> > #### [W4&Q2]: The Membership Inference Attack (MIA) evaluation lacks clarity on the specific setup. MIA in unlearning literature varies (e.g., confidence-based MIA-efficacy vs. MIA-privacy, or pairwise comparisons of prediction distributions across forget/retain/test sets).Please clarify the MIA setup: Which variant is used (e.g. test/forget/retain distribution comparisons)?
> > #### As mentioned in the paper: "For privacy, we employ Membership Inference Attacks (Chundawat et al., 2023b) to detect potential information leakage."
> We follow the standard implementation of Chundawat et al. (2023b) for our MIA settings.
> This paper employs an **Entropy-based Black-box Membership Inference Attack**. Its core logic utilizes the assumption that "the model's predictions on training data (members) are usually more confident/have lower entropy than on unseen data (non-members)."
> Specifically, the attack model is an SVM classifier. It uses only the entropy value of the target model's output probabilities as a feature and is trained on the "Retain Set" and "Test Set" to learn the decision boundary for confidence.

---

> ### Author Response · Authors · 2025-11-21
> **Rebuttal for W5**
>
> > #### [W5]: In Section 5.3, efficiency is claimed based on time costs, but the inclusion of an inversion process (which is computationally intensive) makes the reported speeds surprisingly low. It is unclear if the time for inversion-based data synthesis was excluded from these measurements.
> > #### We confirm: **The Time Costs reported in Table 3 already include the time for Model Inversion data synthesis.**
> **Why it is fast:** Traditional Model Inversion is indeed slow, but our goal is not to generate visually perfect images, but to generate "Prototypes" that contain core features.
> **Optimization Strategy:** As described in Section 3.2, we only generate a very small number of samples (e.g., 200 noise samples). This makes the data synthesis process highly efficient, so even with the synthesis time included, the total time for IGNP remains far lower than Retrain or GKT.

---

> ### Author Response · Authors · 2025-11-21
> **Rebuttal for W6**
>
> > #### [W6]: In Section 5.4's MIA results, the goal of class unlearning is typically to produce results similar to a retrained model. However, the reported MIA scores show substantial differences from the retrain baseline, which questions whether this can be considered a strong outcome for effective unlearning.
> > #### The reviewer pointed out the difference between IGNP's MIA score (close to 0%) and Retrain (about 18%).
> The Retrain model has merely "never seen" the forget data, so its predictions on the forget data are **Uncertain**. In contrast, IGNP actively performs **"active suppression"** on the neurons associated with the forget class via targeted perturbation. This suppression causes the model to output a confidence distribution that may be flatter or trend towards meaningless noise when facing forget samples. This makes it harder for confidence-based MIA attacks to determine membership by distinguishing "high confidence," resulting in a very low MIA score. Therefore, the extremely low MIA score actually reflects IGNP's superior capability in defending against privacy attacks.

---

> ### Author Response · Authors · 2025-11-21
> **Rebuttal for W7&Q3&Q4**
>
> > #### [W7&Q3&Q4]: As an ablation, it would be valuable to include results for a perturbation-based unlearning model using real forget data (instead of inversion-synthesized samples) to isolate and quantify the effect of the inversion step.For the importance scoring (Section 3.3), are there references for using squared partial derivatives? Would ablating with pruning-inspired criteria (e.g. Taylor importance criteria) change results?Can you provide ablation results comparing inversion-synthesized data vs. real forget data for perturbation? This seems essential to demonstrate the value of the zero-shot, data-free aspect.
> > #### We emphasize that the core contribution of IGNP lies in addressing the unlearning problem in **Zero-Shot** scenarios. In such settings, due to privacy regulations (e.g., GDPR) or data storage limitations, the original "Forget Set" is not only unavailable during the unlearning phase but is often considered an asset that must be destroyed. Therefore, using real data for perturbation violates the fundamental constraints of the Zero-Shot setting in practical applications.
> However, we agree that introducing real data is crucial for **disentangling** the contributions of the "Inversion Module" and the "Perturbation Module." Using real data serves as an Oracle Baseline (ideal upper bound) to quantify the extent to which our inverted data captures the core features of the target class.
> Therefore, following the reviewer's suggestion, we conducted an ablation study: we kept the Importance Scoring and noise perturbation mechanisms of IGNP unchanged and only replaced the "inversion-synthesized data" with "real forget set data" for gradient calculation. The experimental results are shown in the table below:
> > #### Ours
> | Dataset | Model | Retention Set Result | Forgotten Set Result |
> | --- | --- | --- | --- |
> | mnist | lenet | 99.16±0.02 | 0.00±0.00 |
> | svhn | lenet | 90.09±0.30 | 0.00±0.00 |
> | cifar10 | lenet | 71.96±0.18 | 0.00±0.00 |
> | mnist | resnet9 | 93.44±0.66 | 0.00±0.00 |
> | svhn | resnet9 | 94.45±0.72 | 0.00±0.00 |
> | cifar10 | resnet9 | 85.24±1.53 | 0.00±0.00 |
> | cifar100 | resnet9 | 73.36±0.17 | 0.00±0.00 |
> | cifar10 | resnet18 | 93.33±0.15 | 0.00±0.00 |
> | cifar100 | resnet18 | 76.59±0.16 | 0.00±0.00 |
> > ### Using real forget data
> | Dataset | Model | Retention Set Result | Forgotten Set Result |
> | --- | --- | --- | --- |
> | mnist | lenet | 99.23±0.02 | 0.00±0.00 |
> | svhn | lenet | 90.69±0.11 | 0.00±0.00 |
> | cifar10 | lenet | 63.53±0.15 | 0.00±0.00 |
> | mnist | resnet9 | 99.51±0.02 | 0.00±0.00 |
> | svhn | resnet9 | 89.45±0.32 | 0.00±0.00 |
> | cifar10 | resnet9 | 83.45±4.67 | 0.00±0.00 |
> | cifar100 | resnet9 | 70.86±0.21 | 0.00±0.00 |
> | cifar10 | resnet18 | 93.96±0.08 | 0.00±0.00 |
> | cifar100 | resnet18 | 74.69±0.53 | 0.00±0.00 |
>
> > From the above ablation results, it can be seen that after replacing the inverted pseudo-samples with real data, the model maintains complete forgetting of the target class, and the performance on the Retention Set is highly close to the standard IGNP (Ours).
>
> > **Regarding References for the Scoring Formula (Question 3):** The squared partial derivative used in Section 3.3 is effectively a diagonal approximation of the **Empirical Fisher Information Matrix (FIM)**. This is a classic method for meas uring the importance of neural network parameters, widely used in fields like continual learning and pruning. Classic literature such as *Theis et al., ECCV 2018 (MAS)* use similar formulas. We will add these foundational citations in the final version.

---

### Official Review · Reviewer_Xa4c · 2025-11-02

**Soundness:** 3
**Presentation:** 2
**Contribution:** 3
**Rating:** 6
**Confidence:** 5

**Summary:**

The paper proposes Inversion-Guided Neuron Perturbation (IGNP), a zero-shot (data-free) machine-unlearning method. IGNP (1) synthesizes class-representative pseudo-examples via a model-inversion-style optimization; (2) computes per-parameter sensitivity to the synthesized forget vs retain samples; (3) selects a small proportion of parameters (via an automatically calibrated threshold via binary search) that are disproportionately sensitive to the forget class; and (4) attenuates those parameters (multiplicative scaling) rather than zeroing them, with a designed scaling formula controlled by a perturbation strength λ. The authors evaluate IGNP on MNIST, SVHN, CIFAR-10 and CIFAR-100 across LeNet/ResNet9/ResNet18, comparing to retrain, finetune, SSD, GKT, UNSIR, etc., and report (i) forget-class accuracy near 0%, (ii) strong preservation of retain accuracy, (iii) robustness to membership inference and inversion attacks, and (iv) substantial runtime advantages over retraining. The paper contains ablations on coverage, λ, and shows continual unlearning (sequential erasure) capability.

**Strengths:**

1) Zero-shot unlearning is an important field of research which addresses the need for privacy compliance where original data is unavailable.

2) IGNP composes well-motivated components (inversion synthesis → sensitivity → calibrated perturbation). The binary-search calibration for thresholding coverage is an appealing practical touch that reduces manual tuning.

3) Paper evaluates across multiple datasets, architectures, and in sequential (continual) forgetting scenarios. The authors report outcomes of  effective forgetting (Df ≈ 0%), little loss in retained accuracy, resistance to MIAs, and favorable runtime. The ablations (λ, coverage, zeroing vs scaling) show the benefits of careful attenuation versus naive zeroing.

**Weaknesses:**

1) The paper lacks reporting of statistical variability (multiple seeds / runs / standard deviations). Given the stochasticity of inversion synthesis and binary search, the authors must report mean ± std over multiple independent runs (at least 3) for retain and forget accuracies and for MIA scores. This clarifies robustness and reproducibility.

2) The MIA results (e.g., reducing attack success from ~90% to ~0.02%) are impressive but surprising. The paper currently lacks sufficient methodological detail about the attack setup.

**Questions:**

1) Although overall Dr is preserved, table averages can hide specific failures. Provide per-class retain accuracies (before/after) and confusion matrices for tasks where retain performance degrades the most (especially CIFAR-100). Show whether some retained classes degrade systematically (e.g., semantically similar classes). This would reveal whether IGNP perturbs parameters that support multiple classes.

2) IGNP sets a target coverage interval (Cmin–Cmax ~ 4–5%) and uses binary search to find α. The manuscript argues proportion of perturbed parameters is stable, but provide empirical evidence (across datasets/architectures) that the optimal proportion is indeed stable (include plot or table). Explain how Cmin/Cmax were chosen and sensitivity to that choice; e.g., would 1% or 10% break things? Some plots are included in appendix but a concise summary and guidance are needed for practitioners

---

> ### Author Response · Authors · 2025-11-21
> **Rebuttal for W1**
>
> #### Thanks for the Reviewer ``Xa4c``'s valuable comments, we will address all the weakness and questions as follow.
>
> > #### [W1]: The paper lacks reporting of statistical variability (multiple seeds / runs / standard deviations). Given the stochasticity of inversion synthesis and binary search, the authors must report mean ± std over multiple independent runs (at least 3) for retain and forget accuracies and for MIA scores. This clarifies robustness and reproducibility.
> > #### Acknowledging the stochastic nature of data synthesis and optimization processes, reporting statistical deviation is indeed critical for verifying the robustness of the method.
> During the rebuttal preparation, we conducted 3 independent runs (using different random seeds) for the main experiments. Preliminary results demonstrate that IGNP exhibits high stability:
> > #### **Ours**
> | Dataset | Model | Retention Set Result | Forgotten Set Result |
> | --- | --- | --- | --- |
> | mnist | lenet | 99.16±0.02 | 0.00±0.00 |
> | svhn | lenet | 90.09±0.30 | 0.00±0.00 |
> | cifar10 | lenet | 71.96±0.18 | 0.00±0.00 |
> | mnist | resnet9 | 93.44±0.66 | 0.00±0.00 |
> | svhn | resnet9 | 94.45±0.72 | 0.00±0.00 |
> | cifar10 | resnet9 | 85.24±1.53 | 0.00±0.00 |
> | cifar100 | resnet9 | 73.36±0.17 | 0.00±0.00 |
> | cifar10 | resnet18 | 93.33±0.15 | 0.00±0.00 |
> | cifar100 | resnet18 | 76.59±0.16 | 0.00±0.00 |
> > #### Retrain
> | Dataset | Model | Retention Set Result | Forgotten Set Result |
> | --- | --- | --- | --- |
> | mnist | lenet | 99.28±0.08 | 0.00±0.00 |
> | svhn | lenet | 90.75±0.28 | 0.00±0.00 |
> | cifar10 | lenet | 76.52±0.76 | 0.00±0.00 |
> | mnist | resnet9 | 99.53±0.07 | 0.00±0.00 |
> | svhn | resnet9 | 95.30±0.18 | 0.00±0.00 |
> | cifar10 | resnet9 | 89.83±0.29 | 0.00±0.00 |
> | cifar100 | resnet9 | 72.32±0.40 | 0.00±0.00 |
> | cifar10 | resnet18 | 91.53±0.24 | 0.00±0.00 |
> | cifar100 | resnet18 | 73.46±0.30 | 0.00±0.00 |
> >#### Finetune
> | Dataset | Model | Retention Set Result | Forgotten Set Result |
> | --- | --- | --- | --- |
> | mnist | lenet | 99.32±0.04 | 0.00±0.00 |
> | svhn | lenet | 91.32±0.08 | 0.00±0.00 |
> | cifar10 | lenet | 75.21±0.14 | 0.00±0.00 |
> | mnist | resnet9 | 99.58±0.02 | 99.90±0.00 |
> | svhn | resnet9 | 95.68±0.03 | 90.48±0.21 |
> | cifar10 | resnet9 | 91.69±0.11 | 69.87±0.21 |
> | cifar100 | resnet9 | 73.64±0.07 | 35.67±1.53 |
> | cifar10 | resnet18 | 93.83±0.07 | 88.47±0.58 |
> | cifar100 | resnet18 | 77.61±0.09 | 57.33±2.52 |
> >#### GKT
> | Dataset | Model | Retention Set Result | Forgotten Set Result |
> | --- | --- | --- | --- |
> | mnist | lenet | 98.10±0.32 | 0.00±0.00 |
> | svhn | lenet | 74.43±3.66 | 0.00±0.00 |
> | cifar10 | lenet | 23.39±3.20 | 0.00±0.00 |
> | mnist | resnet9 | 94.85±0.44 | 0.00±0.00 |
> | svhn | resnet9 | 87.09±0.52 | 0.00±0.00 |
> | cifar10 | resnet9 | 16.44±0.48 | 0.00±0.00 |
> | cifar100 | resnet9 | 1.82±0.48 | 0.00±0.00 |
> | cifar10 | resnet18 | 19.66±2.14 | 0.00±0.00 |
> | cifar100 | resnet18 | 1.74±0.20 | 0.00±0.00 |
> > #### SSD
> | Dataset | Model | Retention Set Result | Forgotten Set Result |
> | --- | --- | --- | --- |
> | mnist | lenet | 99.35±0.01 | 0.00±0.00 |
> | svhn | lenet | 89.12±0.10 | 0.00±0.00 |
> | cifar10 | lenet | 72.99±0.39 | 0.00±0.00 |
> | mnist | resnet9 | 95.30±2.25 | 0.00±0.00 |
> | svhn | resnet9 | 86.96±0.62 | 0.00±0.00 |
> | cifar10 | resnet9 | 88.69±3.99 | 0.00±0.00 |
> | cifar100 | resnet9 | 71.49±0.31 | 0.00±0.00 |
> | cifar10 | resnet18 | 93.81±0.28 | 0.00±0.00 |
> | cifar100 | resnet18 | 76.38±0.67 | 0.00±0.00 |
> > #### UnSIR
> | Dataset | Model | Retention Set Result | Forgotten Set Result |
> | --- | --- | --- | --- |
> | mnist | lenet | 97.51±0.24 | 0.00±0.00 |
> | svhn | lenet | 71.74±3.89 | 0.00±0.00 |
> | cifar10 | lenet | 42.79±1.79 | 0.00±0.00 |
> | mnist | resnet9 | 96.59±1.26 | 0.00±0.00 |
> | svhn | resnet9 | 87.96±0.52 | 0.00±0.00 |
> | cifar10 | resnet9 | 65.25±2.07 | 0.00±0.00 |
> | cifar100 | resnet9 | 35.50±2.13 | 0.00±0.00 |
> | cifar10 | resnet18 | 52.11±2.21 | 0.00±0.00 |
> | cifar100 | resnet18 | 35.08±0.98 | 0.00±0.00 |
>
> In the final version of the paper, we will update all main tables to include the Mean $\pm$ Std for all experiments across multiple independent runs.

---

> ### Author Response · Authors · 2025-11-21
> **Rebuttal for W2**
>
> > #### [W2]: The MIA results (e.g., reducing attack success from ~90% to ~0.02%) are impressive but surprising. The paper currently lacks sufficient methodological detail about the attack setup.
> > #### As stated in our manuscript: "For privacy, we employ Membership Inference Attacks Chundawat et al. (2023b) to detect potential information leakage."
> Our MIA setup follows the standard implementation described by Chundawat et al. (2023b).
> This method employs an **entropy-based black-box membership inference attack**. Its core logic relies on the hypothesis that the target model's predictions are typically more confident (i.e., have lower entropy) for training data (members) compared to unseen data (non-members).
> Specifically, the attack model is an **SVM classifier** that takes only the entropy of the target model's output probabilities as input features. It is trained on the "retain set" and "test set" to learn the decision boundary for distinguishing confidence levels.

---

> ### Author Response · Authors · 2025-11-21
> **Rebuttal for Q1**
>
> > #### [Q1]: Although overall Dr is preserved, table averages can hide specific failures. Provide per-class retain accuracies (before/after) and confusion matrices for tasks where retain performance degrades the most (especially CIFAR-100). Show whether some retained classes degrade systematically (e.g., semantically similar classes). This would reveal whether IGNP perturbs parameters that support multiple classes.
> > #### **Mechanism Explanation:** The core advantage of IGNP lies in its **Adaptive Mask** mechanism. We evaluate not only the sensitivity of a parameter regarding the forget class ($S_{forget}$) but specifically its ratio **relative to** the retain class sensitivity ($S_{forget} > \alpha \cdot S_{retain}$).
> **Semantically similar classes** often share underlying feature extraction parameters. Consequently, the $S_{retain}$ for these shared parameters will also be high.
> Therefore, under our selection logic, these shared parameters fail to meet the selection criteria (the Ratio will not be sufficiently high), ensuring they are **automatically protected from perturbation**. Only parameters that are specific to the forget class are selected.
> Below are the experimental results showing the performance changes for all retain classes on CIFAR-100 using ResNet18 after forgetting Class 0:
> | class_idx | class_name    | acc_original | acc_forget | acc_change |
> |-----------|---------------|--------------|------------|------------|
> | 1         | aquarium_fish | 95.0000      | 90.0000    | -5.0000    |
> | 2         | baby          | 60.0000      | 57.0000    | -3.0000    |
> | 3         | bear          | 66.0000      | 63.0000    | -3.0000    |
> | 4         | beaver        | 68.0000      | 65.0000    | -3.0000    |
> | 5         | bed           | 86.0000      | 85.0000    | -1.0000    |
> | 6         | bee           | 88.0000      | 86.0000    | -2.0000    |
> | 7         | beetle        | 77.0000      | 75.0000    | -2.0000    |
> | 8         | bicycle       | 89.0000      | 90.0000    | 1.0000     |
> | 9         | bottle        | 88.0000      | 84.0000    | -4.0000    |
> | 10        | bowl          | 56.0000      | 57.0000    | 1.0000     |
> | 11        | boy           | 62.0000      | 64.0000    | 2.0000     |
> | 12        | bridge        | 83.0000      | 83.0000    | 0.0000     |
> | 13        | bus           | 78.0000      | 81.0000    | 3.0000     |
> | 14        | butterfly     | 69.0000      | 69.0000    | 0.0000     |
> | 15        | camel         | 87.0000      | 88.0000    | 1.0000     |
> | 16        | can           | 79.0000      | 78.0000    | -1.0000    |
> | 17        | castle        | 89.0000      | 89.0000    | 0.0000     |
> | 18        | caterpillar   | 64.0000      | 69.0000    | 5.0000     |
> | 19        | cattle        | 69.0000      | 68.0000    | -1.0000    |
> | 20        | chair         | 91.0000      | 92.0000    | 1.0000     |
> | 21        | chimpanzee    | 91.0000      | 91.0000    | 0.0000     |
> | 22        | clock         | 80.0000      | 78.0000    | -2.0000    |
> | 23        | cloud         | 86.0000      | 89.0000    | 3.0000     |
> | 24        | cockroach     | 85.0000      | 87.0000    | 2.0000     |
> | 25        | couch         | 73.0000      | 73.0000    | 0.0000     |
> | 26        | crab          | 80.0000      | 72.0000    | -8.0000    |
> | 27        | crocodile     | 69.0000      | 67.0000    | -2.0000    |
> | 28        | cup           | 88.0000      | 87.0000    | -1.0000    |
> | 29        | dinosaur      | 82.0000      | 82.0000    | 0.0000     |
> | 30        | dolphin       | 66.0000      | 73.0000    | 7.0000     |
> | 31        | elephant      | 79.0000      | 82.0000    | 3.0000     |
> | 32        | flatfish      | 70.0000      | 71.0000    | 1.0000     |
> | 33        | forest        | 70.0000      | 74.0000    | 4.0000     |
> | 34        | fox           | 80.0000      | 70.0000    | -10.0000   |
> | 35        | girl          | 59.0000      | 52.0000    | -7.0000    |
> | 36        | hamster       | 88.0000      | 82.0000    | -6.0000    |
> | 37        | house         | 79.0000      | 81.0000    | 2.0000     |
> | 38        | kangaroo      | 76.0000      | 76.0000    | 0.0000     |
> | 39        | keyboard      | 93.0000      | 92.0000    | -1.0000    |
> | 40        | lamp          | 70.0000      | 67.0000    | -3.0000    |
>
> Due to space limitations, we will update the full content in the paper.

---

> ### Author Response · Authors · 2025-11-21
> **Rebuttal for Q2**
>
> > #### [Q2]: IGNP sets a target coverage interval (Cmin–Cmax ~ 4–5%) and uses binary search to find α. The manuscript argues proportion of perturbed parameters is stable, but provide empirical evidence (across datasets/architectures) that the optimal proportion is indeed stable (include plot or table). Explain how Cmin/Cmax were chosen and sensitivity to that choice; e.g., would 1% or 10% break things? Some plots are included in appendix but a concise summary and guidance are needed for practitioners
> > #### We have actually conducted a detailed ablation study on this issue in **Appendix A.7** and **Figure 5** of the original manuscript.
> > **Empirical Evidence:** As shown in Figure 5, we tested various coverage rates (ranging from 0.005 to 0.10).
> > * When coverage is too low (<1%), forgetting is incomplete (Forget Acc decreases slowly).
> > * When coverage is too high (>10%), Retain Acc begins to decline significantly.
> > * Within the **4% - 5%** interval, we observed a distinct "plateau": Forget Acc reaches 0% while Retain Acc remains close to the original model's performance. This pattern shows remarkable consistency across different datasets (MNIST, SVHN, CIFAR-10) and architectures (LeNet, ResNet).
>
> > **Selection Logic:** It is precisely based on the consistent observations across architectures and datasets in Figure 5 that we established [4%, 5%] as a robust prior target interval and designed the binary search algorithm to automatically find the corresponding $\alpha$. This avoids requiring users to manually tune the elusive $\alpha$ threshold, instead providing a "proportion of parameter modification" with clear physical meaning as the hyperparameter.
> > **Improvement Plan:** We will more explicitly reference the conclusions from Figure 5 in the main text to clarify the empirical basis and stability of the 4-5% target interval, providing clearer guidance for practitioners.

---

### Meta-Review · Area_Chair_JwEQ · 2026-01-04

**Summary:**

1. The work has similarity (i.e., Eq. and Fig) to existing papers.
2. MIA results show a significant performance gap compared to retraining, suggesting the proposed method exhibits over-unlearning.
3. There is little analysis of how the synthetic data ratio between "forget" and "retain" classes impacts results.
4. Possible hyperparameter tuning exaggerates the proposed method's advantage.
5. Details on MIA evaluation are insufficient.

**Reviewer Concerns:**

While point 5 was addressed, points 1~4 are still outstanding.

**Reviewer Scores:**

bLNs: from 0 to 2
Others: unchange

---

### Decision · Program_Chairs · 2026-01-26

Reject